# Technical note: High accuracy weighing micro-lysimeter system for long-term measurements of non-rainfall water inputs to grasslands

Andreas Riedl[1], Yafei Li[1], Jon Eugster[2], Nina Buchmann[1], Werner Eugster[1]

[1]Department of Environmental Systems Science, ETH Zurich, Zurich, 8092, Switzerland
[2]School of Mathematics, The University of Edinburgh, Edinburgh, EH9 3FD, UK

*Correspondence to*: Andreas Riedl (andreas.riedl@usys.ethz.ch)

**Abstract.** Non-rainfall water (NRW), defined here as dew, hoar frost, fog, rime and water vapor adsorption, might be a relevant water source for ecosystems, especially during summer drought periods. These water inputs are often not considered in ecohydrological studies, because water amounts of NRW events are rather small and therefore difficult to measure. Here

we present a novel micro-lysimeter (ML) system and its application which allows to quantify very small water inputs from NRW during rainfree periods with an unprecedented high accuracy of $\pm 0.25$ g, which corresponds to $\pm 0.005$ mm water input. This is possible with an improved ML design paired with individual ML calibrations in combination with high-frequency measurements at 3.3 Hz and an efficient low-pass filtering to reduce noise level. With a set of ancillary sensors, the ML system furthermore allows differentiating between different types of NRW inputs: dew, hoar frost, fog, rime and the

combinations among these, but also additional events when condensation on leaves is less probable, such as water vapor adsorption events. In addition, our ML system design allows to minimize deviations from natural conditions in terms of canopy and soil temperatures, plant growth and soil moisture. This is found to be a crucial aspect for obtaining realistic NRW measurements in short-statured grasslands. Soil temperatures were higher in the ML compared to the control, thus further studies should focus to improve the thermal soil regime of ML. Our ML system has proven to be useful for high-

accuracy, long-term measurements of NRW on short-statured vegetation like grasslands. Measurements with the ML system at a field site in Switzerland showed that NRW input occurred frequently with 127 events over 12 months, with a total NRW input of 15.9 mm. Drainage water flow of the ML was not measured, therefore the NRW inputs might be conservative estimates. High average monthly NRW inputs were measured during summer months, suggesting a high ecohydrological relevance of NRW inputs for temperate grasslands.

# 1 Introduction

Non-rainfall water (NRW) inputs, defined here as dew, hoar frost, fog, rime and water vapor adsorption, provide water to plants. These different inputs form under different environmental conditions: Dew forms on plant surfaces when the temperature of the surface drops below the dewpoint temperature of the adjacent air (Beysens, 2018; Monteith, 1957), whereas dew forming directly on soil surfaces is

rarely observed (Agam and Berliner, 2004; Ninari and Berliner, 2002). In addition, hoar frost is frozen dew, which forms at temperatures below 0 °C. Fog droplets form on condensation nuclei (activated aerosol particles) in the atmosphere when water vapor concentration reaches saturation, whereas rime is supercooled fog in contact with a surface (e.g. vegetation) at a temperature below 0 °C. Water vapor adsorption occurs on hygroscopic surfaces, which can lower saturation vapor pressure and thus lead to
adsorption, despite the fact that temperatures are still above dewpoint temperature (Agam and Berliner, 2006; McHugh et al., 2015).

NRW inputs are a water source for plants during dry periods and can thus have a significant influence on plant water relations by increasing plant water status (Boucher et al., 1995; Kerr and Beardsell, 1975; Wang et al., 2019; Yates and Hutley, 1995). Plant water status is a widely used measure in plant
physiology for assessing plant water stress. It incorporates the amount of water in plants and its energy status (Jones, 2006). NRW inputs can increase the amount of water in plants (Limm et al., 2009; Munné-Bosch and Alegre, 1999) and increase thereby the plant water status, which lowers plant water stress. Plants can take up NRW via the leaves, termed foliar water uptake (Berry et al., 2014; Eller et al., 2013; Slatyer, 1960), or via the roots (Wang et al., 2019). NRW is brought to the rhizosphere by drip-
off from leaves and stems (Dawson, 1998), or by dew formation and/or fog droplet interception and impaction on soils (Agam and Berliner, 2006; Kaseke et al., 2012; Uclés et al., 2013). Moreover, NRW can also reduce water loss (1) by suppressing transpiration (Aparecido et al., 2016; Gerlein-Safdi et al., 2018; Ishibashi and Terashima, 1995; Waggoner et al., 1969), induced by clogged stomata (Gerlein-Safdi et al., 2018; Vesala et al., 2017); (2) by reducing the vapor pressure deficit (Ritter et al., 2009) in
the boundary layer between leaves and the atmosphere; and (3) by decreasing canopy temperatures because of evaporative cooling during re-evaporation of NRW inputs (Thornthwaite, 1948). The energy from incoming solar radiation is partially used for the phase transition from liquid water to water vapor, which thereby alleviates potential heat stress of the plants. Moreover, canopy temperature may decrease due to an increase in surface albedo (Eugster et al., 2006; Minnis, 1997), when more light is reflected as
long as the surface is wet. Thus, NRW inputs can substantially change water relations and micro-environmental conditions of plants.

Despite these significant effects of NRW on plants, NRW inputs are the least studied component in ecohydrology (Wang et al., 2019), because NRW inputs are difficult to quantify (Groh et al., 2018; Jacobs et al., 2006; Kidron and Starinsky, 2019). High accuracy measurement instrumentation, which
simulates natural conditions, e.g. in terms of surface properties, while minimizing disturbances, is required to capture the comparatively small water inputs. There exists no international agreement on a reference standard instrumentation system for NRW measurements (Chen et al., 2005; Groh et al., 2018). Over the last decades, different measurement systems were developed (see Kidron and Starinsky, 2019). Lysimeter (LM) and micro-lysimeter (ML) systems simulate natural conditions well (Ninari and
Berliner, 2002) and are therefore considered as accurate and reliable NRW measurement methods (Ninari and Berliner, 2002; Richards, 2004; Uclés et al., 2013). Hence, they became the most commonly used methods over the last decades (Kidron and Starinsky, 2019). LM differ from ML by their much larger size, although there is no well-defined size threshold that indisputably allows to separate LM from ML (6 to 25 cm in diameter and 3.5 to 25 cm in depth).

The main drawback of large ML for NRW studies is the trade-off between weighing capacity and weighing accuracy. The weighing capacity of LM and ML is determined by their load cell capacity: the higher the weighing capacity, the lower the weighing accuracy.

Most ML systems were developed for application in arid regions to measure NRW inputs to soils and sand. ML systems for temperate regions may have different requirements, because quantification of
NRW inputs on vegetation requires a sufficient ML size for natural plant (root) growth. ML with shallow depth and small radius can alter normal plant (root) growth, because of insufficient space availability. This characteristic makes them unsuitable for long-term NRW studies on vegetation with a high demand for root space. Furthermore, natural soil–atmosphere water exchange might be altered by shallow depth of the ML in some ecosystems. While limited rainfall retention capacity of ML is not a
problem for NRW quantification, the potential prevention of upward direct water flow due to capillary rise from deeper soil layers or the groundwater body cannot be neglected (Evett et al., 1995), because it replenishes plant available water in the rooting zone. Likewise, the energy budget of small ML can be severely affected by its insufficient depth (Kidron and Kronenfeld, 2017; Ninari and Berliner, 2002).

All LM and ML are disconnected from the surrounding soil and therefore can exhibit a more efficient
heat loss via nocturnal long-wave radiative cooling (Kidron and Kronenfeld, 2017). To accurately measure NRW inputs on short-statured vegetation it is thus crucial that the canopy temperature of the ML vegetation equals the canopy temperature in its surrounding (control). This is especially true for dew formation, hoar frost and water vapor adsorption events. Higher temperatures of ML canopies would lead to underestimated NRW amounts, while lower temperatures would lead to overestimated
NRW amounts (Kidron and Kronenfeld, 2017). Consequently, measuring NRW inputs reliably needs to take these effects into account.

The goal of this study was to design and test an automated long-term ML system for NRW quantification to grasslands during dry (unsaturated soil) and rainfree periods, that overcomes drawbacks of existing small ML systems in terms of hampered plant growth and altered canopy and soil
temperatures as compared to the control (surrounding area). The main objectives of our study were to:

(1) develop a ML system with high accuracy that overcomes existing drawbacks of size vs. accuracy and that does not hinder plant growth and minimises ML temperature differences as compared to its surroundings.
(2) design a ML system that allows differentiating between different NRW inputs, here defined as
dew, hoar frost, fog, rime as well as water vapor adsorption events during dry and drought conditions, and
(3) to test for long-term suitability of the ML system in the field and to quantify the share of NRW of the mean annual precipitation.

## 2 Material and Methods

### 2.1 Field site Früebüel

Field work for this study was carried out at Früebüel (CH-FRU), a long-term Swiss FluxNet field site in Switzerland (Pastorello et al., 2020; Zeeman et al., 2010). The site is a permanent grassland located on a mountain plateau in the Canton of Zug, Switzerland (47°06'57.0" N, 8°32'16.0" E) at an elevation of 982 m a.s.l.. The annual mean temperature is 7.8 °C (years 2005 to 2019), the annual mean rainfall is 1232 mm (SD = ± 372 mm). The site is moderately intensively managed with two to four management events per year, usually a combination of mowing and grazing, depending on vegetation growth (Imer et al., 2013). The dominant species are common ryegrass (*Lolium multiflorum*), meadow foxtail (*Alopecurus pratensis*), cocksfoot grass (*Dactylis glomerata*), dandelion (*Taraxacum officinale*), buttercup (*Ranunculus sp.*) and white clover (*Trifolium repens*) (Sautier, 2007). The soil at the site is a silt loam mixture (56% silt, 37% sand, 7% clay), with a bulk density of $1.12 \pm 0.03$ g cm$^{-3}$ and an organic C content of $4.4 \pm 0.2\%$ (Stiehl-Braun et al., 2011). The main rooting horizon is within the top 20 cm of soil, with a high root density in the top 11 cm (Stiehl-Braun et al., 2011). A location map and an aerial photograph of the site can be found in Appendix A.

The site is equipped with an agrometeorological station, comprising a temperature and a relative humidity sensor (CS215, Campbell Scientific Inc., Logan, USA) placed in an actively aspired radiation shield, a cup anemometer with a wind vane (A100R and W200P, Vector Instruments, North Wales, UK), all installed at a height of 1.15 m, and a 3D anemometer (R3-50, Gill Instruments Ltd., Lymington, UK) installed at a height of 1.80 m. Moreover, the site is equipped with a tipping bucket rain gauge (15188H, Lambrecht meteo GmbH, Goettingen, Germany) and a networked digital camera (NetCam SC, StarDot Technologies, Buena Park, CA, USA). Furthermore, a leaf wetness sensor (PHYTOS 31, Meter Group AG, Munich, Germany) that mimics thermodynamic and radiative properties of a leaf, is installed horizontally at a height of 30 cm, to measure close or in the canopy of the grassland vegetation. A visibility sensor (MiniOFS, Optical sensors Sweden AB, Gothenburg, Sweden) is installed at a height of 1 m to capture shallow radiation fog and rime events.

### 2.2 Methods

The ML system was composed of three individual ML with additional sensors. The three ML were placed in a row at 1.45 m intervals. The design of the ML system is presented in Section 2.2.1 – 2.2.2. Further information about the installation process (including photographs), data processing and storage can be found in the Appendix. A description of the installation procedure and the soil monolith preparation can be found in Appendix B. How data were collected, stored and delivered can be found in Appendix C. The description of the load cell data low-pass filtering can be found in Appendix D.

### 2.2.1 ML design

A ML consisted of an inner part (Fig. 1a) and an outer part (Fig. 1b, item (a), in what follows referenced as Fig. 1b:a). The outer part (Fig. 1b:a) was made by a cylindrical PVC-U tube (VINK Schweiz GmbH,

Dietikon, Switzerland; 45 cm outer diameter x 42 cm height, 44.64 cm inner diameter) with an open top and a closed bottom. The bottom was closed with a PVC-XT disk (VINK Schweiz GmbH, Dietikon, Switzerland; 46 cm diameter, 0.3 cm thick), which was welded with a PVC-U welding rod to the cylindrical tube for waterproof closure. The outer part protected the inner part (Fig. 1b:b–q) from confounding factors like soil pressure, infiltrating water and biota. The core elements of the inner part were a cylindrical pot (Fig. 1b:b), filled with a soil monolith (for simplicity called ML pot within this paper) containing the original grass sward. The ML pot was made of a cylindrical PVC-U tube (VINK Schweiz GmbH, Dietikon, Switzerland; 25 cm outer diameter x 25 cm height, 24.8 cm inner diameter), of which the bottom was closed with a PVC-XT disc (VINK Schweiz GmbH, Dietikon, Switzerland; 26 cm diameter, 0.3 cm thick) that was welded in the same way as the outer part. The ML pot was mounted by means of three custom made sockets (Fig. 1b:c) on a weighing platform (Fig. 1b:d–g), secured with machine screws. The weighing platform consisted mainly of three parts, the load plate (Fig. 1b:d), a load cell (Fig. 1b:e), and a base plate (Fig. 1b:f). The load plate was made of aluminum (AlSi1MgMn, 29 cm diameter, 1 cm thick), likewise the base plate (35 cm diameter, 1 cm thick). Between the load plate and the base plate, a PW15AHY temperature-compensated load cell with 20 kg capacity (HBM, Darmstadt, Germany) was mounted. To allow bending of the load cell, two rectangular spacing washers (Fig. 1g, 2.5 x 3.1 cm, 0.1 cm thick) were mounted between load cell and load plate, and between load cell and base plate. To mount the load cell and the spacing washers to the load plate and the base plate, two countersunk head screws were used. The weighing platform was standing on three equidistant adjustable support feet (Fig. 1b:h, M6×1 machine screws, 15.5 cm height) integrated in the base plate. This allowed to level the weighing platform, which is important for accurate load cell measurements. A counter nut above the base plate (Fig. 1b:i) fixed the position of the weighing platform.

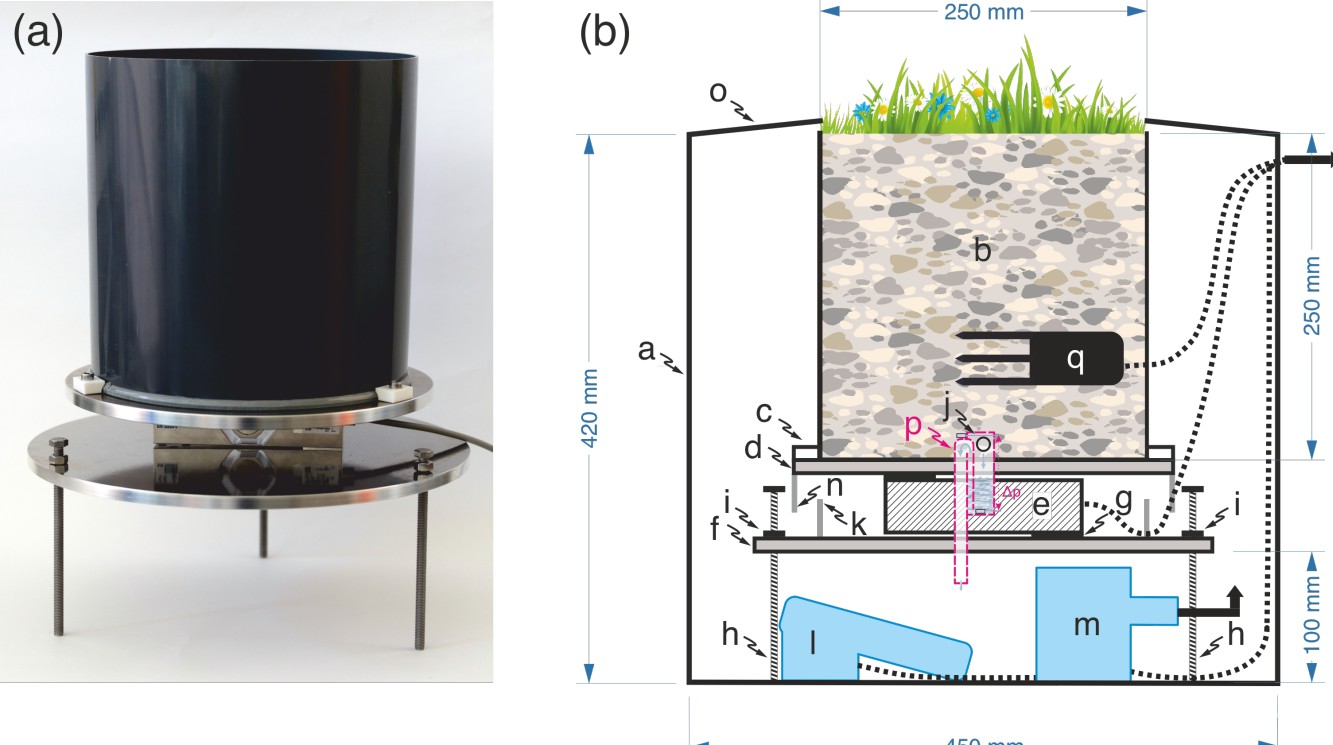

Fig. 1. Inner part of the ML (a) and schematic drawing of ML design (b) with: a) outer part, b) ML pot, c) socket, d) load plate, e) load cell, f) base plate, g) spacing washer, h) adjustable support feet, i) counter nut for adjustable support feet, j) cover lid, k) water guide, l) water and dirt protection, m) float switch, n) bilge pump, o) soil moisture and temperature sensor, p) drainage-water outlet, and q) optional sensor or drop counter to quantify drainage for applications that do not specifically target drought conditions.

### 2.2.2 Drainage water flow

To avoid stagnating water inside of ML pots, a passive drainage water flow path was made. The drainage-water was guided away from the load cell to a reservoir to protect the load cell from suspended matter. Suspended matter can be carried along with drainage water and could impede the function of the load cell by blocking the load cell bending. Drainage water beyond soil field capacity was allowed to flow out from the bottom of the ML pot via drainage-water outlets. Three drainage-water outlets (Fig. 1b:j; 0.8 cm diameter) were drilled equidistantly into the lateral side of the ML pot as close as possible to the bottom. The drainage-water outlets were protected with a metal mesh to prevent erosion of ML soil during heavy rainfall events. Excessive water could follow a passive drainage path from the top of the load plate, guided by a water guide (Fig. 1b:n; 3 cm height, 0.4 cm thick), to the base plate. From the base plate water could flow to an approximately 10 cm high reservoir below the base plate. If the collected water in the reservoir exceeded a certain threshold, a float switch (Fig. 1b:l; Fujian Baida Pump, Fuan, China) gave a signal to a bilge pump (Fig. 1b:m; Fujian Baida Pump, Fuan, China) that

pumped the water away from the ML system (schematically shown with an arrow in Fig. 1b) via a
flexible tube (2 cm inner diameter). The load cell was protected from drainage water flow by a
rectangular water and dirt protection (Fig. 1b:k, PVC XT, 25 cm x 10.5 cm, 4 cm height). It was glued
at the base plate around the load cell and made watertight with silicon.

Rainfall could enter also in the gap between the ML pot and the outer part of the ML system. To
minimize this water collection, a cover lid (Fig. 1b:o) made of a PVC-XT ring (47 cm outer diameter,
26 cm inner diameter) was constructed. The cover lid had an inclination of 7° towards the outside. This
was done by putting the cover lid in a heated oven at 90 °C and then pressing it towards a custom-made
wooden fit with the desired form, till it had cooled down. The slanted cover lid resulted in a preferred
water flow towards the surrounding and thereby prevented water flow towards the inside of the ML
system. Furthermore, it protected the ML pot from incident solar radiation, also minimizing potential
heating effects. Wiring of the load cell, the float switch, the bilge pump as well as the soil temperature
and moisture sensors were bundled and led out close to the top of the outer part of the ML system
(schematically shown with an arrow in Fig. 1b).

In the design as used here, i.e. to quantify NRW inputs during rainfree periods, drainage water was
allowed to freely drain from the ML pots. Thus, rainfall periods had to be excluded from analysis (see
section 2.2.3). However, to use the ML system during and shortly after rainfall periods, it is
recommended to add an additional sensor (Fig. 1b:p) to quantify drainage water flow (see Appendix E).
For applications without such an additional sensor, it should be kept in mind that, depending on soil
type, up to 41.5 hours after intensive rainfall that saturated the soil monolith completely drainage water
losses can occur (see Fig. F1 and Table F1).

**2.2.3 Calculation of NRW amounts and differentiation of NRW inputs**

We differentiated six types of NRW events with ML and ancillary sensors, i.e., (1) dew only, (2) hoar
frost only, (3) fog only, (4) rime only, (5) combined dew and fog events, and (6) combined hoar frost
and rime events. During all six event types, a mass increase was expected on the ML. The NRW
amounts (NRW$_{amount}$) were calculated using equation (1):

$$NRW_{mass} = \begin{cases} ML_{max1m} - ML_{min1m}, & precip = 0 \text{ mm} \\ 0, & precip > 0 \text{ mm} \end{cases}$$ (1)

where $ML_{max1m}$ is the maximum value of the one-minute mean ML mass (all three ML values averaged
every minute) over a time period of 24 hours (from 12:00 to 12:00 UTC), $ML_{min1m}$ is the minimum
value of the one-minute mean ML mass over the same time period. The resulting NRW$_{mass}$ (in grams)
was then converted to mm. If rainfall occurred during an analyzed 24-hour period, that period was
excluded, except when the rain event occurred directly after the NRW input event. Rain events were
determined by the rain gauge measurements at the site. Time periods with a snow cover as determined
visually from digital images were not considered in the analysis. To distinguish between different types
of NRW inputs, we used the information from all ancillary sensors. Often dew and fog or hoar frost and
rime occurred in combination, e.g. after sunset, dew formation occurred, when the atmosphere cooled
further down till the atmosphere got highly saturated, fog started to form. We termed such events

combined dew and fog events, or hoar frost and rime events, respectively. The leaf wetness sensor was used to sense condensation (during dew only and hoar frost only events), NRW droplet interception and impaction (during fog, rime, combined dew and fog, combined hoar frost and rime events), and to sense an absence of condensation (during events when less condensation is expected to occur, e.g. water vapor adsorption or dew formation on soil). The visibility sensor was used to distinguish between events with reduced visibility below 1000 m (fog, rime events), and events without reduced visibility (dew only, hoar frost only events). To distinguish between fog and rime events from dew and hoar frost events, the temperature sensor of the nearby agrometeorological station was used. When temperature dropped below 0 °C, NRW inputs were attributed to rime and hoar frost.

## 2.2.4 Load cell calibration and determination of accuracy

In this study, weighing accuracy denotes the difference between the measured mass (determined with a ML) and the control (calibrated mass). Precision reflects the reliability of the measurements, and it specifies to what extent the experiment can be repeated. On the other hand, resolution is the smallest distinguishable unit for an observable change in mass and thus determines the upper limit of precision. For NRW studies, high accuracy is indispensable, which requires instruments with high resolution paired with high precision.

Calibration runs for ML and the determination of the accuracy of the measurements were performed in a laboratory with closed windows and doors to avoid any influence of turbulence on load cell readings. Raw data were filtered as described in Appendix D during load cell calibration of the ML. A two-point calibration was performed on every single ML using calibration mass. For mass increases up to 500 g, calibration mass complying with the OIML F1 standard (Mettler Toledo, Greifensee, Switzerland) were used. The maximum permissible error of these calibration mass is $\pm$ 2.5 mg. For mass increases of 1000 g, custom made mass of steel were used. Their mass was determined on a laboratory scale (XS4002S DeltaRange, Mettler Toledo, Switzerland) which was calibrated and certified for determining mass up to 4.1 kg with an accuracy of $\pm$ 0.01 g. First, a zero-point calibration was carried out, then the span was set to 15045.2 g, as this was the approximate mass which most moist ML pots had. The offset from the zero-point calibration was used together with the span calibration value in the code running on the microcontroller. The absolute accuracy of the load cells was tested on 2[nd] April 2019, by loading calibration mass on the weighing platform, in the range of 0 kg to 19.5 kg. The mass was increased stepwise by 500 g. The maximum mass was set to 19.5 kg to avoid an overload damage of the load cell. Three repetitions were performed. A linear regression was performed in order to assess the relationship between target mass and load cell mass. Moreover, a relative calibration was performed on 7[th] April 2019. We investigated the accuracy of a load cell with relative mass changes. A base mass, ranging from 10 kg to 19.5 kg, was loaded on the weighing platform, then a 100 g calibration mass was added to the base mass. Accuracy of relative mass changes was determined with three replications. To test accuracy also under field conditions, we regularly performed a loading/unloading experiment after Nolz et al. (2013), by loading 5 to 10 g calibration mass on the ML and noting the mass before and after the loading. Because masses can be calibrated with certified standards as was done here, we use the term accuracy in this context, which goes beyond (relative) precision.

### 2.2.5 Evaluation of the effects of ML size on plant growth, canopy temperatures and soil moistures and temperatures

Plant growth in the ML system was evaluated by comparing individual plant heights in the ML pots versus the control (surrounding). Plant heights were measured from ground level to maximum standing height. Plant heights of *Trifolium pratense*, *Plantago major* and *Rhinanthus alectorolophus* were measured at CH-FRU on 26 July 2019, with three replications per species and treatment (ML pot, control). To test for a statistically significant difference between plant heights of ML pots and the control (surrounding) we used a *t*-test (n=3). To compare canopy temperatures of ML and the control (surrounding) during a NRW input period, we used a thermal camera (testo 882, Testo AG, Lenzkirch, Germany), with a thermal sensitivity of ±0.05 °C. Thermal infrared images were taken from 18:27 to 05:15 (UTC) of ML vegetation and of the control (surrounding) at CH-FRU during a dew night on 24 to 25 June 2019. Thermal images of the control (surrounding) were taken in a distance of ca. 100 cm from the ML system, to exclude any potential influences of the ML system on its immediate surrounding. To compare thermal images of the ML surface with the control, we compared the variance (F-test). Data were bootstrapped to reduce sample size from > 30k to 30 samples using the scikit-learn machine learning package of Python (Pedregosa et al., 2011). Soil moisture and temperature data of ML pots and the control (surrounding) were retrieved by soil temperature and moisture sensors (Fig. 1b:q; 5TM, Meter Group AG, Munich, Germany), installed at a soil depth of 15 cm. As a control, one additional sensor was placed outside the ML system at the same depth in the surrounding. We measured over a period from beginning of May till mid October 2019. Soil moisture data were compared as water filled pore space (WFPS). WFPS was used to make soil moisture values better comparable, by minimizing the effects of soil texture, e.g. different gravel content, that might be present in close proximity of the sensors. Higher or lower gravel content could bias soil saturation. WFPS was calculated relative to a saturation point (100%), which was reached, when the soil was heavily saturated with water after long and intensive rainfall. To test if the difference of WFPS values of ML pots and the control (surrounding) stayed constant over time, we used a cointegration test after Engle and Granger (1987), which can be used to test for co-movement of two non-stationary variables. To test if the WFPS time series were non-stationary, we used an Augmented Dickey-Fuller (ADF) test. To perform all statistical tests, we used the Statsmodels package (Skipper et al., 2010) of Python.

## 3 Results

### 3.1 Accuracy of the ML system

Three replications showed an almost perfect linear correlation ($R^2$=0.9999) between target mass and load cell mass. Target mass was retrieved from the microcontroller after data filtering (see Appendix D). Data with a resolution of 0.1 g were used. The root mean square errors (RMSE) for comparisons of target mass to load cell mass of three replications were 0.43, 0.47 and 0.36 g, respectively. The standard

error (SE) of the parameter estimates of three replications were ± 0.13, ± 0.14 and ± 0.11 g, respectively.

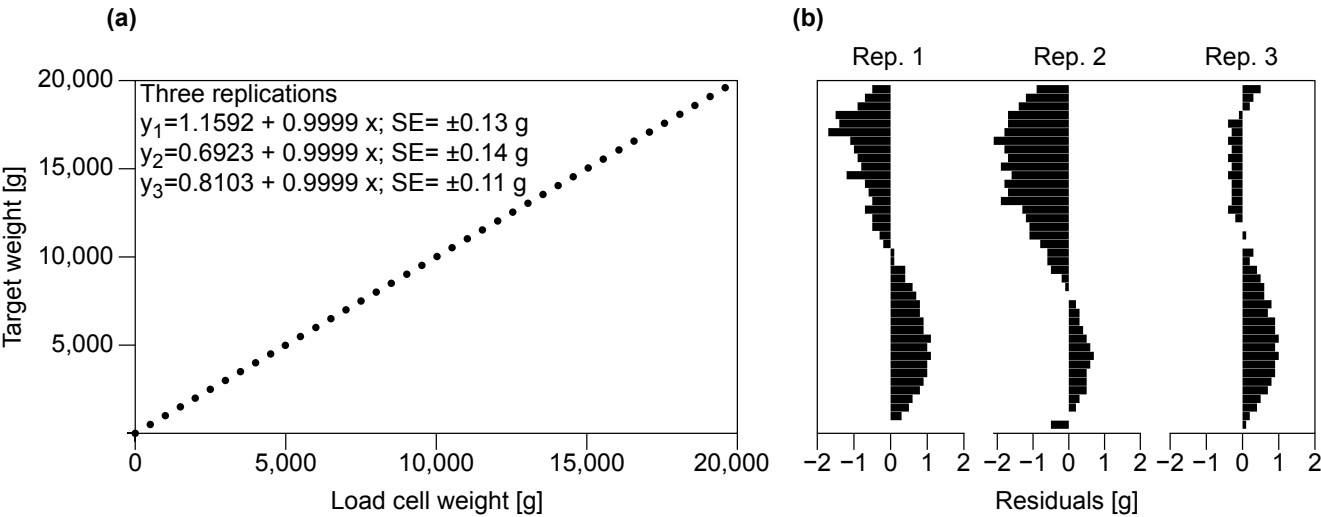

**Fig. 2 (a) Absolute calibration of a load cell placed in a weighing platform. Three replications (overlapping data points) are shown with SE of the intercept. (b) The residuals from the target mass of three replications (Rep. 1 to 3) were in the range of ± 2 g.**

NRW inputs occur during events with a finite time period, thus for NRW input studies, the relative change in mass from start to end of that time period is of interest. A 100 g change with the given ML size translated to a change of 2 mm water input. The residuals were in the range of ± 0.25 g or ± 0.005 mm equivalent water input, which represents the accuracy of the ML system.

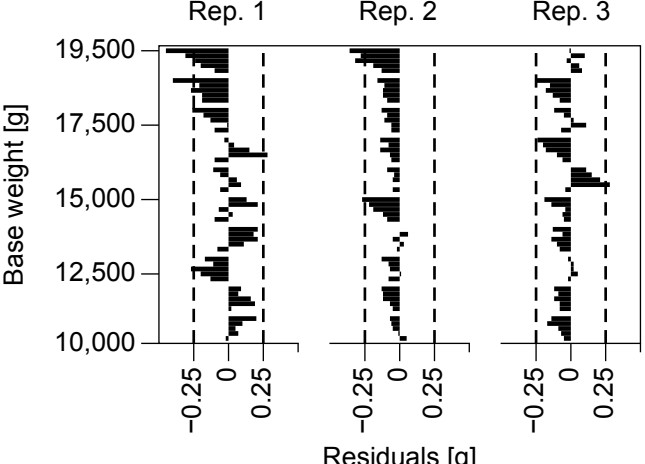

**Fig. 3 Residuals of three replications (Rep. 1 to 3) with relative mass changes of 100 g.**

A zero-point offset calibration combined with data filtering (see Appendix D) gave us not only a more
accurate zero-point offset, but also a more accurate span value. An accurate span value reduced fluctuating values from load cell readings and gave us stable measurements when mass changed over time. The precision was determined by repeatedly loading and unloading calibration mass on the weighing platform for three times and noting the difference to test for repeatability. The precision was $\pm$ 0.28 g, equivalent to $\pm$ 0.005 mm water input. With a base mass over 18.5 kg, the precision was slightly
lower, with $\pm$ 0.45 g equivalent to $\pm$ 0.009 mm water input. The digital resolution of the ML system was 0.01 g, which corresponds to 0.0002 mm equivalent water input, and is thus two orders of magnitude better than the physical resolution provided by our ML system. Regular loading/unloading experiments after Nolz et al. (2013) showed deviations in the range between $\pm$ <0.1 g ($\pm$ <0.002 mm) and $\pm$ 0.4 g ($\pm$ 0.008 mm), and thereby confirmed high accuracy also under field conditions. Thus, the data acquisition
of the ML system was accurate enough to provide high accuracy.

**3.2 Differentiation among different types of NRW inputs**

Our ML system allowed differentiating among different types of NRW events when the ML measurements were combined with ancillary sensors. During a combined dew and fog event (Fig. 4a), we measured an increase in mass on the ML, an increase in leaf wetness (uncalibrated sensor voltage),
while visibility was partially below 1000 m (intermittent fog event). During a dew only event, we measured an increase in mass on the ML, besides increased leaf wetness, while visibility stayed above 1000 m throughout the event (Fig. 4b). During a potential water vapor adsorption event, there was only an increase in mass on the ML, whereas no condensation occurred on the leaf wetness sensor, while the visibility stayed well above 1000 m (Fig. 4c). Wind speed remained low (< 1 m s$^{-1}$) during the whole
potential water vapor adsorption event. Mass increases on the ML could be attributed to hoar frost if air temperature was below 0 °C or to rime during events with reduced horizontal visibility <1000 m and temperatures below 0 °C. The highest water gain of the NRW input events shown in Fig. 4 was 0.4 mm and originates from the combined dew and fog event; the water input from the dew only event was 0.2 mm, and the lowest water input with 0.06 mm came from the potential water vapor adsorption event.

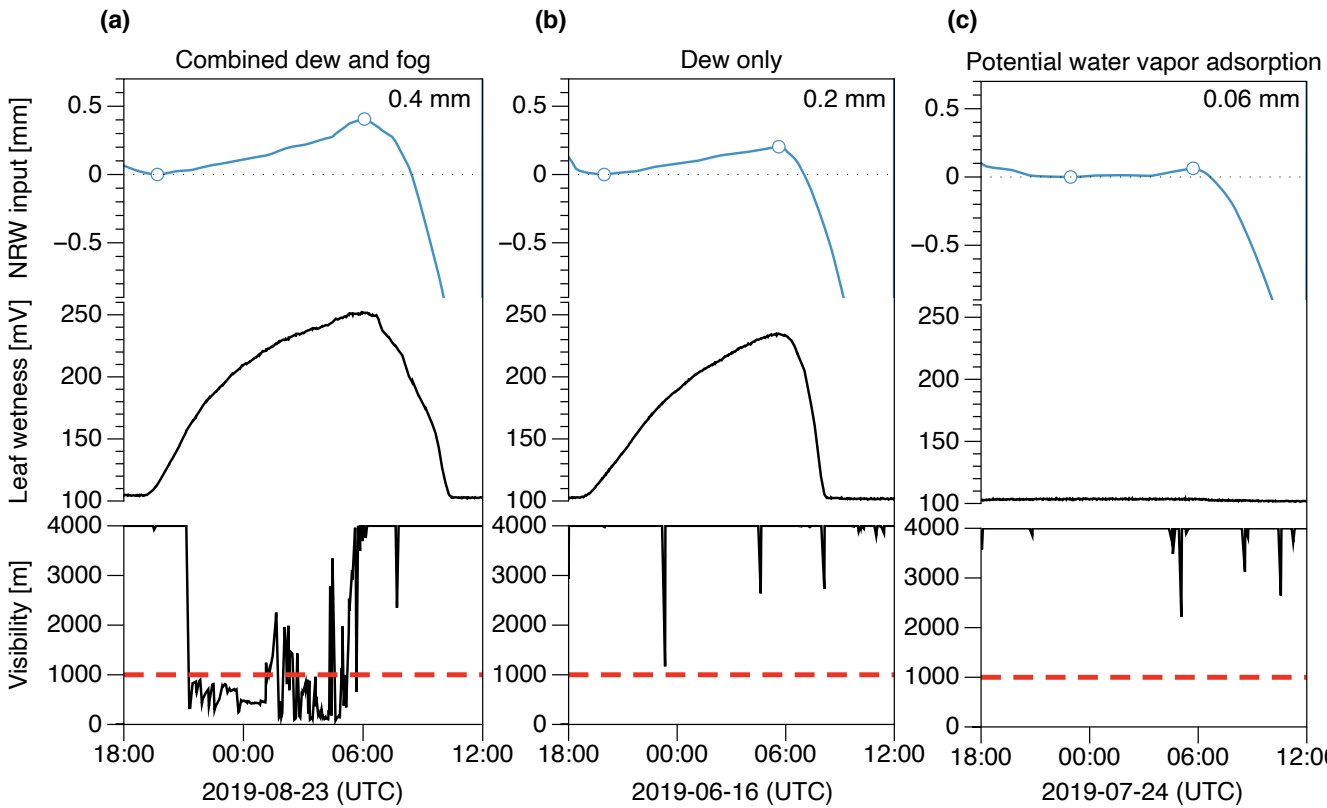


**Fig. 4. Differentiation of different NRW input events with the ML system and ancillary sensors: (a) Combined dew and fog event; (b) Dew only event; (c) Potential water vapor adsorption event. The black dashed line indicates the zero line. The red dashed line is the threshold for fog events with a visibility < 1000 m. Visibilities > 4000 m were reported as 4000 m. Blue circles indicate start and end of NRW input events.**



**Table 1. Cross table to indicate different criteria for differentiation among different NRW events. The '+' sign indicates the presence, whereas the '-' sign indicates the absence of a certain factor. All NRW events lead to increase of ML mass, ancillary sensors of leaf wetness, visibility and temperature are needed to differentiate between NRW events.**

| NRW event type | ML mass increase | Leaf wetness | Visibility < 1000 m | Temperature < 0 °C |
|---|---|---|---|---|
| Dew | + | + | − | − |
| Hoar frost | + | + | − | + |
| Fog | + | + | + | − |
| Rime | + | + | + | + |
| Combined dew and fog | + | + | + | − |
| Combined hoar frost and rime | + | + | + | + |
| Potential water vapor adsorption | + | − | − | − |

### 3.3 Influence of ML system design on plant canopy temperature

Canopy temperature did not differ significantly (*t*-test, $p > 0.05$, $n = 30$) between ML vegetation and control (Fig. 5a, b). The standard deviation of temperature data between ML surface and the control was < 0.5 °C throughout the observation period. The variance of canopy temperature between the ML vegetation and the control was not statistically significant different (F-test, $p > 0.05$, $n = 30$). Soil temperature in the ML pot 1 was higher than in the control plot at the beginning of the dew formation period (Fig. 5c), but equaled control soil temperatures towards the end. Dew formation started at 18:53 and ended at 06:07 UTC (Fig. 5d). Dew water input was 0.24 mm, showcased for ML 1, even though dew formation occurred during that night on all three ML installed at the site.

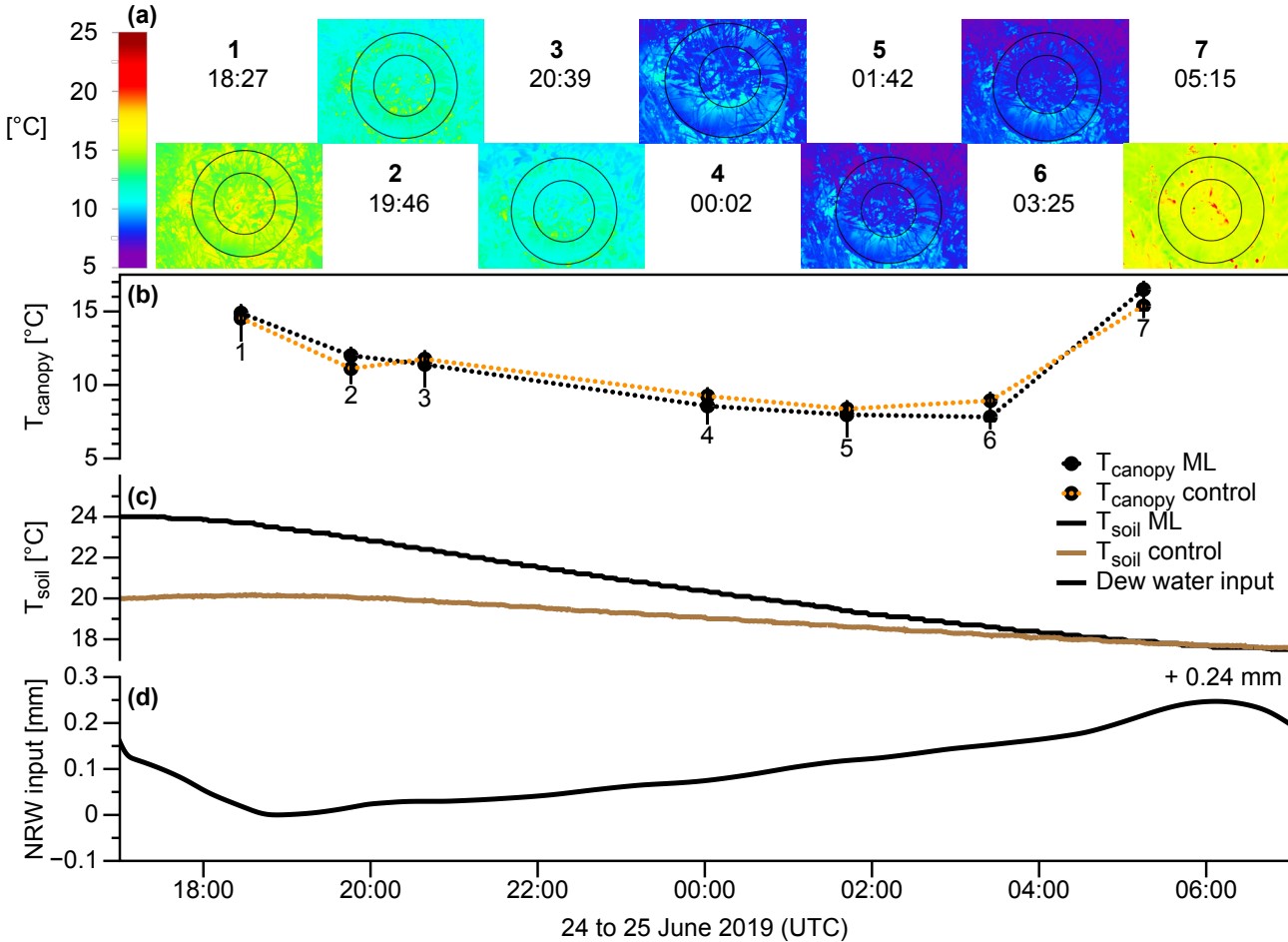

Fig. 5. Canopy temperatures (a, b), soil temperatures (c), and NRW input (d) of ML1 and the control (surrounding area) at CH-FRU during 24 to 25 June 2019. Time of day (HH:MM) is given in UTC time. The thermal infrared images (a) show the ML pot (small circle) with the cover lid (between small circle and big circle) and the surrounding (outside of big circle) during selected time points (1–7) of a dew night. Image size is ca. 75×75 cm. To compare ML pot temperatures to temperatures of the surrounding, separate images were taken in a distance of ca. 100 cm (images not shown here) with a size of ca. 75×75 cm, to
exclude any potential influence of the ML on its approximate surrounding.

## 3.4 Influence of ML system design on plant growth

Plant heights of *Trifolium pratense*, *Plantago major* and *Rhinanthus alectorolophus* did not differ between ML pots and the control (*t*-test, p > 0.05, n = 3), also variability did not differ (F-test, p > 0.05, 375 n = 3). Additional measurements of mean and maximum vegetation height on 14 August 2019 showed also no statistically significant difference (*t*-test, *p* > 0.05, n = 3; data not shown).

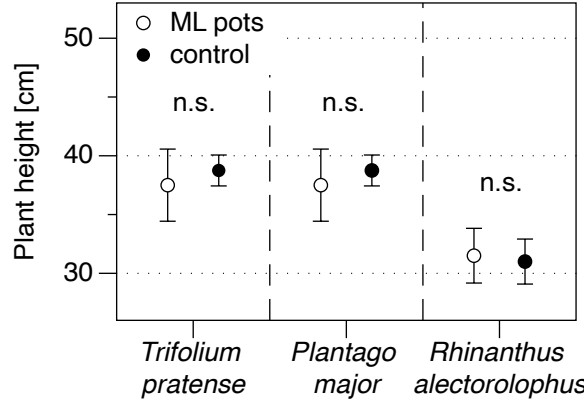

**Fig. 6. Comparison of plant height of three plant species at CH-FRU (measured on 26 May 2019) growing in ML pots versus the same species growing in the open field (control). Error bars are showing standard errors (n = 3), n.s. stands for no statistically significant difference.**

## 3.5 Influence of ML system design on soil moistures and temperatures

WFPS data of ML pots 1, and ML pot 2 were very similar, and closely matched the control (Fig. 7a). WFPS values of ML pot 3 showed a higher dynamic, but closely followed the temporal pattern of the control and ML pots 1 and 2. The differences between WFPS of ML pots and the control were constant over time (Engle-Granger two step cointegration test; $p < 0.05$). This indicates that soil moisture data of ML pots and the control were in general not significantly different. However, during a prolonged no-rainfall period in summer (Fig. 7a, marked with red box), WFPS of ML pots decreased faster in comparison to the control. Since lower soil moisture values can result in a lower heat capacity of the soil, we assessed whether lower WFPS values inside ML pots may have an influence on soil temperature during non-rainfall periods (Fig. 7b).

Soil temperature of ML pot 1 and the control (Soil temperature in the surrounding) (Fig. 7b) showed the same increasing trend, while deviation of WFPS of ML pots from the control (Fig. 7a, marked in red) increased with time (same pattern as of ML pot 1 was also evident on ML pot 2 and ML pot 3, data not shown). From this we conclude that soil temperatures inside ML pots during the most relevant hours of the day when dew forms (during the night before sunrise) were not strongly influenced by a lower water content and its resulting lower heat capacity. Nocturnal temperature minima almost perfectly agreed between ML pot 1 and the control, while the daily temperature range of ML pot 1 was double compared to the control (Fig. 7b). Over the prolonged no-rainfall period, the hourly mean soil temperature deviations of ML pot 1 from the control ranged between –0.14 °C around sunrise and 2.57 °C in the later afternoon (Fig. 7c). Over the period from May-October 90 % of nocturnal one-minute soil temperature deviations (sunset–sunrise) were lower than 2.90 °C, 50 % were lower than 0.69 °C.

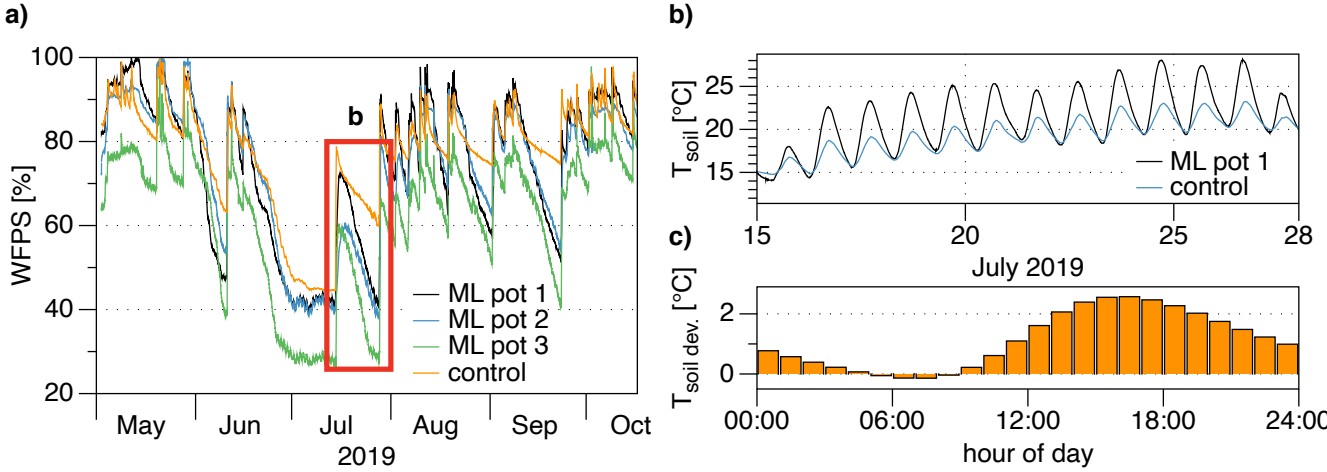

**Fig. 7. (a) Comparison of WFPS (based on soil moisture measured at 15 cm depth) inside the ML pots versus the control from beginning of May till mid of October 2019 at CH-FRU;( b) Soil temperature from ML pot 1 at CH-FRU during a non-rainfall period in July (marked with red box in panel a); (c) Soil temperature deviations of ML pot 1 from the control by hour of day during the same period as marked in panel a and used in panel b.**

## 3.6 NRW inputs over one year

There were a total of 127 NRW input events at CH-FRU over one year (2nd May 2019 12:00 UTC to 2nd May 2020 11:59 UTC; Fig. 8). The frequency of the events can be found in Table 2. Eleven NRW events were observed when leaf wetness remained low, potentially indicating water vapor adsorption events or dew formation on soil. Potential water vapor adsorption events occurred during two time periods: period 1 in July 2019, period 2 in April 2020. During period 1, a single potential water vapor adsorption event occurred, whereas during period 2 ten such events occurred. During both periods rainfall was low, ten days before the event in period 1 the cumulative rainfall was only 9.6 mm, in period 2 the cumulative rainfall between 14 March, the last bigger rainfall event with 12.3 mm, and 23 April was only 13.7 mm. The soil moisture during both potential water vapor adsorption periods was rather low, with WFPS of ca. 45 %. This indicates a potential water vapor gradient from the atmosphere to the soil, favorable for water vapor adsorption. The cumulative NRW input over 12 months was 15.9 mm, which corresponds to roughly 1% of the 1580 mm annual precipitation collected during the third warmest year in Switzerland since weather recordings started in 1864 (MeteoSchweiz, 2020).

**Table 2. Number counts of events with its associated NRW input by type, and percentage of the total NRW input during the observation period of 12 months at CH-FRU.**

| Number count of events | NRW type | NRW input (mm yr⁻¹) | NRW input (mm d⁻¹) | Percentage of total NRW input (%) |
|---|---|---|---|---|
| 85 | dew | 10.23 | 0.12 | 64.23 |
| 21 | hoar frost | 1.92 | 0.09 | 12.05 |
| 13 | Combined dew and fog | 2.69 | 0.21 | 16.89 |
| 5 | fog | 0.9 | 0.18 | 5.67 |
| 2 | Hoar frost and rime | 0.15 | 0.08 | 0.95 |
| 1 | rime | 0.03 | 0.03 | 0.22 |

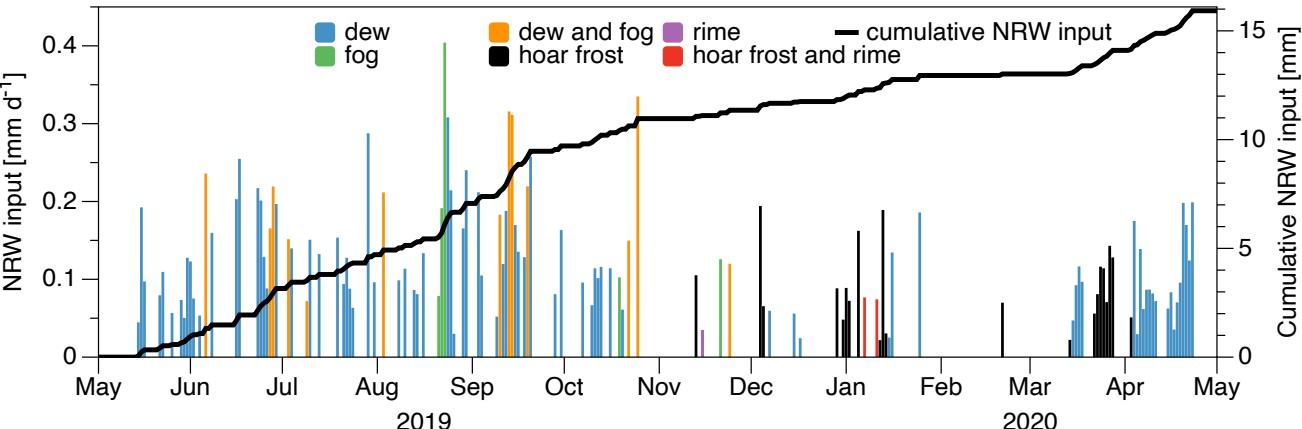

**Fig. 8. Daily NRW inputs at CH-FRU over one year, starting on 2nd May 2019 till 2nd May 2020. The blue bars indicate NRW events with their corresponding NRW input per day. Different colours indicate different types of NRW inputs. The black line indicates the cumulative NRW input over one year. The annual total NRW input was 15.9 mm, about 1% of total precipitation during this time.**

The mean NRW input over all events was 0.12 mm, with the highest single input of 0.4 mm by a fog event, and the lowest input of 0.021 mm by a hoar frost event. On a monthly basis, the months with highest NRW inputs were September with 2.64 mm, August with 2.35 mm, and June with 2.32 mm. The cumulative NRW input from May until September was 9.7 mm. At the monthly scale, NRW inputs can be remarkable: in April 2020, the month with the least rainfall (51.8 mm), the contribution of NRW

input to the monthly hydrological input was 3.5%. The average monthly NRW input was highest in September with 0.088 mm, when the nights were longer than in summer, and thus the probability for NRW inputs was increasing with the duration of the night. However, observed average monthly NRW

inputs ranked second and third in terms of amount in June and August when nights were much shorter than in September. The relationship between NRW input as a function of actual NRW input duration (Fig. 9) was not very strong, but when durations were binned into ten bins of equal widths, a clear trend of increasing NRW inputs with increasing NRW input duration emerged. Because no NRW input is expected if the duration of NRW input is 0 hours, we first started with a square-root regression through the origin, $y = b \cdot \sqrt{x}$, the slope of the fit was $0.042 \pm 0.001$ mm h$^{-1/2}$ (Fig. 9 dotted line, R2 = 0.98, p < 0.001), but for durations > 2 hours it closely corresponded to a conventional linear regression slope of 0.008 ± 0.001 mm h–1 (Fig. 9 black line, R2 = 0.86, p < 0.001; the intercept should be ignored because it has no physical meaning in this context). Despite this rather clear dependence on actual duration of NRW input, there was no significant correlation found between average monthly NRW input duration and potential NRW input duration given by the time between sunset and sunrise (R2 = 0.16, p > 0.1; data now shown).

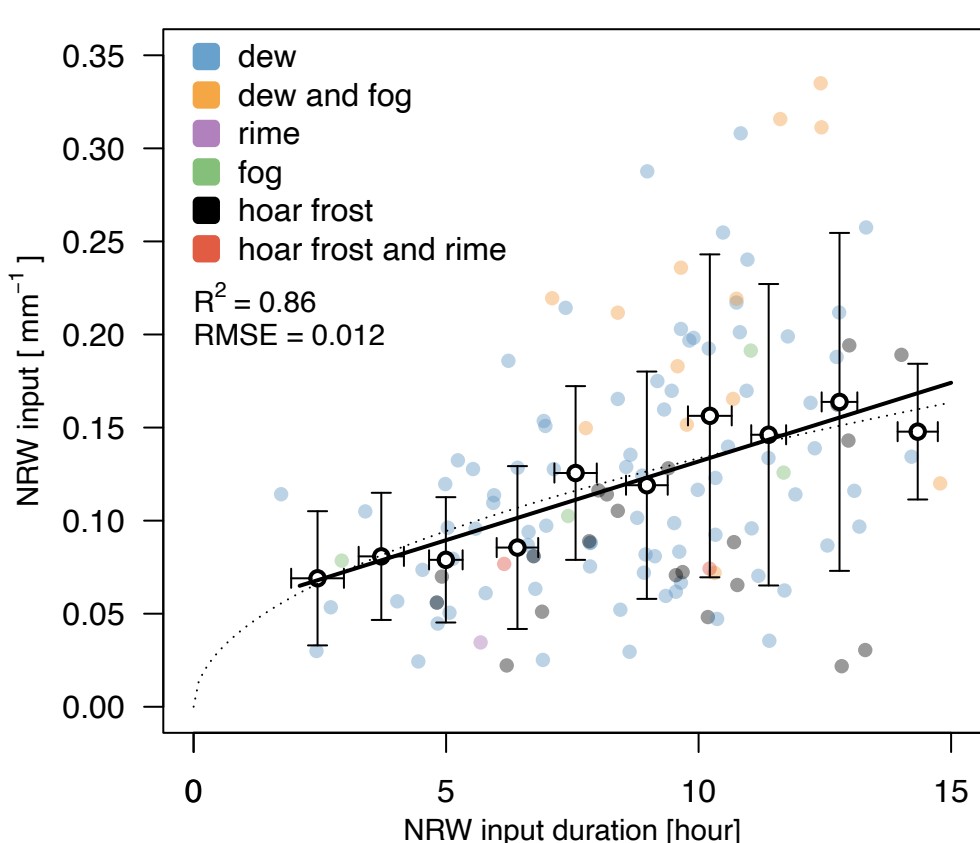

**Fig. 9. The relationship of actual NRW input as a function of actual NRW input duration from 12 months of NRW inputs. NRW inputs were binned to 10 bins of equal width covering the entire data range of the NRW input duration. Horizontal and vertical whiskers indicate the SD of the available data within each bin relative to the respective bin average (open circles). Different**

colours indicate different types of NRW inputs. There is a strong linear relationship ($R^2 = 0.86$, $p < 0.001$) between actual NRW input and actual NRW input duration.

## 4 Discussion

### 4.1 Accuracy of the ML system

The high accuracy of our newly developed ML system allowed capturing even very small NRW events
such as the potential water vapor adsorption event with 0.06 mm shown in Fig. 4c. It was possible to capture NRW events with an accuracy of ± 0.25 g with pots that weigh roughly 15 kg in total. This corresponds to an accuracy of ± 0.005 mm of water inputs. The accuracy would be even higher with a relative mass change less than 100 g (equivalent to 2 mm water input), which is true for most NRW events. The accuracy of our ML system was four orders of magnitude better than reported for many
other studies (see Table 3). Feigenwinter *et al.* (2020) could achieve on average (depending on calibration date) the same accuracy, although with a lower depth of the ML pot (6.5 cm) and a lower weighing capacity (7kg). The high accuracy of our ML system was achieved by a combination of factors, such as using a state-of-the-art load cell in combination with continuous high frequency data filtering as well as ancillary data. For example, temperature measurements were crucial to differentiate
between hoar frost and dew events and fog and rime events. Ancillary wind measurements could be used to exclude periods with high wind speeds, because high wind could act as a force on ML and increase thereby mass. However, NRW inputs occur during conditions with low wind speed, the probability for dew formation decreases below 5% when wind speeds are smaller than 0.4 m s$^{-1}$ or bigger than 1.9 m s$^{-1}$ (Zhang et al., 2014). Thus, wind is not a big bias source for NRW quantification.
A further factor promoting high accuracy was a load-cell specific calibration. Factory calibration is the same for all load cells of the same model, but when an individual calibration is made, the differences among individual load cells are substantial, and hence highest accuracy always requires a load-cell specific calibration by the user. Construction details that promoted accuracy were the frictionless gap construction between ML pot and cover lid, as well as the three adjustable support feet on which the
weighing platform was centred on the load cell. This is needed because after burial, a ML system may accidentally tip, twist and be thrown out of balance (Uclés et al., 2013). The low-cost microcontroller had enough computing power to continuously process data from multiple sensors, while consuming little energy. Thus, our ML system could also be powered by solar panels. During or after freezing temperature conditions the ML system should be controlled, because expanding water in the reservoir or
the ML pot could break PVC parts of the ML system. However, this did not occur during this study period.

Precision (repeatability of the measurements) of our ML system was ± 0.005 mm equivalent water input. With a base mass over 18.5 kg, the precision was lower, with ± 0.009 mm equivalent water input. However, in the field, ML pots were weighing less than 18.5 kg, even when soil was moist. This
precision was unprecedented, only topped by manual ML weighing on an electronic balance (Jia et al., 2014). Manual weighing is, however, very labor intensive and consequently unsuitable for long-term NRW studies.

The digital resolution (smallest distinguishable unit) of our ML system was 0.0002 mm. This resolution was in the range reported by Uclés *et al.* (2013). Comparison of accuracies, precisions and resolutions with other studies is often hampered, because the distinct terms accuracy, precision and resolution are often misconceived. The load cell capacity of 20 kg in our ML system is relatively large compared to other ML studies. NRW input studies with ML had a load cell capacity in the range from 0.3 kg (Brown et al., 2008), 1.5 kg (Kaseke et al., 2012), 3 kg (Uclés et al., 2013), 6 kg (Maphangwa et al., 2012; Matimati et al., 2013), up to 7 kg (Feigenwinter et al., 2020).







**Table 3. Comparison of accuracies, precisions, and resolutions of ML and LM for NRW studies.**

| Accuracy of ML and LM | Additional information | Reference |
|---|---|---|
| ± 0.005 mm | ML weighing capacity of 20 kg | This study |
| ± 0.005 mm (mean) | Accuracy ranged from ± 0.001 mm to ± 0.02 mm depending on calibration date. ML weighing capacity of 7 kg | Feigenwinter *et al.* (2020) |
| ± 0.02 mm | ML weighing capacity of 1 kg | Heusinkveld *et al.* (2006) |
| ± 0.03 mm | | Zhang *et al.* (2019) |
| **Precision of ML and LM** | | |
| ± 0.28 g (± 0.005 mm) | | This study |
| ± 0.001 g (± 0.00012 mm) | ML pots were manually weighed on an electronic balance | Jia *et al.* (2014) |
| ± 0.3 g (± 0.008 mm) (mean) | Precision ranged from ± 0.1 g (± 0.002 mm) to ± 1.12 g (± 0.023 mm), depending on calibration date | Feigenwinter *et al.* (2020) |
| ± 20 g (± 0.01 mm to ± 0.04 mm) | For a surface area of 0.5 $m^2$ up to 2 $m^2$ | Meissner *et al.* (2014) |
| **Resolution of ML and LM** | | |
| 0.01 g (± 0.0002 mm) | | This study |
| 0.01 g (± 0.00055 mm) | | Uclés *et al.* (2013) |
| 0.038 g (± 0.0026 mm) | | Kaseke *et al.* (2012) |
| 0.1 g (± 0.0022 mm) | | Maphangwa *et al.* (2012) |
| 0.1 g (± 0.004 mm) | | Agam and Berliner (2004) |
| 1 g and 10 g (± 0.001 mm | Big LM, two different LM | Groh *et al.* (2018) |

| and 0.01 mm) | systems with 1 m$^2$ surface area |
| --- | --- |

## 4.2 Quantification and differentiation among different types of NRW inputs

NRW inputs occurred rather frequently over the entire year of observation (Fig. 8). NRW inputs could be measured on approximately every third day on average. The highest NRW inputs occurred during the months of main grass growth (April–September), indicating a potential hydro ecological relevance. Ancillary sensors allowed differentiation of different NRW inputs. Differentiation among different types of NRW inputs is important for various research disciplines, e.g. the prediction of fog events poses a major challenge for numerical weather prediction for meteorologists (Westerhuis et al., 2020). Thus, it is important to measure the frequency and water inputs of fog events during the whole year.

The use of a visibility sensor allowed us to assess the contribution of fog and rime. A leaf wetness sensor allowed differentiating between events in which condensation occurred (dew, hoar frost) in contrast to events when condensation on leaves was less probable (water vapor adsorption and/or dew formation on soil). Potential water vapor adsorption events occurred during periods with low rainfall, when soil was drying out, which increased the vapor pressure deficit gradient between soil and atmosphere, promoting water vapor adsorption. However, the NRW inputs of the potential water vapor adsorption events were rather low (0.03 – 0.13 mm). Thus, it is not unlikely that a leaf wetness sensor might react slightly different than a true plant leaf, despite the care that was taken to design leaf wetness sensors to match the radiative and thermodynamic properties of plant leaves, and these events were small dew events. Further investigations are needed to clarify if the leaf wetness sensor is suitable to differentiate between dew and water vapor adsorption events. Air temperature measurements from the agrometeorological station were necessary to differentiate between dew vs. hoar frost formation and between fog vs. rime. Rainfall measurements allowed differentiating between NRW events and rainfall events, and a networked digital camera allowed to observe persisting snow cover. The installation of three ML allowed exclusion of possible effects by insects, snails and lizards arriving on or departing from a ML pot. If it is assumed that these animals have no preference for a particular ML pot and thus their arrival and departure is a random process, such effects only contribute to the noise that is filtered out during data filtering, and thus should not bias our NRW input estimates. In deserts or arid regions (with low vegetation cover) additional sensors (e.g. infrared video cameras) would be needed to detect depositing materials like dust and sand that accumulate on the ML over time. The installation of multiple ML further had the advantage that spatial variation in soils, species composition and leaf area could be reduced in comparison to single ML deployments.

## 4.3 Effect of ML size on plant growth, canopy temperatures, soil moisture and soil temperatures

Our ML system had a larger area and a deeper ML pot than most other ML systems developed and used in earlier studies on NRW quantification (Table 4). This allowed unimpaired plant height growth (Fig. 6), representing more natural conditions than many, rather shallow ML systems, an issue crucial for

accurate measurements of NRW inputs to grasses and forbs. We did not find any significant differences in canopy temperatures between our ML pots and of the control (surrounding) (Fig. 5a). Furthermore, we found in general no significant difference in soil moisture between ML and the control (surrounding), only during a prolonged drought period soil moisture values of ML pots were decreasing faster. In this study, this had however no influence on plant standing height because measurements of
plant height (before the drought period) and measurement of overall vegetation height (after the drought period) were not statistically different. However, lower soil moisture during prolonged drought periods can result in reduced evaporation rates and increased water vapor adsorption rates. Furthermore, this can influence plant growth and development. Thus, the ML system can be used to reliably measure NRW inputs as long as the difference in soil moisture during prolonged drought periods does not
influence plant height or canopy architecture. WFPS values of ML pots were in general not higher than the control, suggesting a sufficient drainage by the drainage-water outlets. This is crucial, because saturation at the bottom of ML could lead to oxygen limitation for root growth (Ben-Gal and Shani, 2002). In contrast to Kidron and Kronenfeld (2017), Evett *et al.* (1995) and Ninari and Berliner (2002), we also did not observe substantially lower nocturnal soil temperatures, the time when NRW inputs
actually take place, which is important to avoid an overestimation of dew formation on soils. On the other hand, afternoon and close to sunset soil temperatures of ML pots were higher compared to those in the control (Fig. 7). Thus, potentially, the ML system could underestimate dew formation on soils shortly after sunset, but dew formation on soils is rare (Agam and Berliner, 2004; Ninari and Berliner, 2002), the open soil surface in grasslands is rather small, ideally zero under good management
practices. Higher soil temperatures could underestimate water vapor adsorption, because it lowers the vapor pressure deficit between soil and atmosphere. Therefore, our estimates of NRW inputs on soils should be conservative estimates, given that the slightly elevated temperatures actually do reduce (not increase) NRW inputs on soil inside the ML pots. The higher soil temperatures in the afternoon were not related to a lower water content nor its associated heat capacity. Kidron *et al.* (2016) provided a
possible explanation for the diurnal temperature difference between a ML pot and the control. They termed it a "loose stone effect", the ML pot might act as loose stone, i.e., through the air gap between the ML pot and the outer part of the ML system more efficient longwave radiational cooling can occur in comparison to the bulk soil. However, Ninari and Berliner (2002) found that the lateral soil temperature gradient was small compared to the vertical soil temperature gradient and that wrapping the
ML pots with insulation material did not reduce temperature deviations. We thus think that insufficient ML pot depth has most likely caused the soil temperature alterations observed mainly during daytime when dew formation is absent. Ninari and Berliner (2002) suggested that the minimum ML depth should be the depth at which the temperature is constant during the entire day. For a dry loess soil in the Negev Desert, a sufficient ML pot depth would be 50 cm (Ninari and Berliner, 2002). At CH-FRU, a
ML pot depth of approximately 95 cm would be necessary, in order to have soil temperature gradients over 24-hour periods < 0.5 °C. With a depth of 95 cm, there would be the risk that all the advantages any ML system entails would be lost. Although constructing deeper ML pots would be possible, even with double or triple the current ML pot depth, deeper ML pots would exert more dead mass onto the load cell and would thus decrease load cell accuracy (Kaseke et al., 2012). Overall, ML design is always
a tradeoff between representing the surrounding and feasibility of construction and installation. The ML

system was not constructed with the depth suggested by Ninari and Berliner (2002), however, the aim of this study was to measure NRW inputs to grasslands, for which canopy temperatures are more important. We found only a small difference in canopy temperature between ML and the control. Thus, we conclude that our novel ML design is suitable for quantifying nocturnal NRW inputs on grasses and

forbs reliably and accurately at high temporal resolution.






**Table 4. Size comparison of lysimeters (LM) and micro-lysimeters (ML) developed and used for NRW studies.**

| LM or ML | Depth [cm] | Diameter [cm] | Study object | Locality | Reference |
|---|---|---|---|---|---|
| ML | 25 | 25 | grassland | CH-FRU (Früebüel, Switzerland) | This study |
| LM | 150 | 112 | grassland | Gumpenstein, Rollesbroich (Austria and Germany) | Groh *et al.* (2018) |
| LM | 200 | 112 | cropland *(Zea mays)* | Helmholtz Centre for Environmental Research – UFZ (Germany) | Meissner *et al.* (2007) |
| LM | 265 | 225 | herbaceous vegetation | Dingxi (China) | Zhang *et al.* (2019) |
| ML | 3.5 | 6 | sand dunes | Nizzana, Negev desert (Israel) | Jacobs *et al.* (1999) |
| ML | 3.5 | 6 | undisturbed soil with biological soil crusts | Gurbantunggut desert (China) | Zhang *et al.* (2009) |
| ML | 3.5 | 8.8 | soil | Knersvlakte (South Africa) | Brown *et al.* (2008) |
| ML | 3.5 | 14 | sand | Nizzana, Negev desert (Israel) | Heusinkveld *et al.* (2006) |
| ML | 3.5 | 14 | river sand | Stellenbosch (South Africa) | Kaseke *et al.* (2012) |
| ML | 3.5 | 24 | gypsum soils and lichens | Alexander bay (South Africa) | Maphangwa *et al.* (2012) |
| ML | 3.5 | 24 | dwarf succulents | Quaggaskop, Knersvlakte (South Africa) | Matimati *et al.* (2013) |

| | | | | | |
|---|---|---|---|---|---|
| ML | 6.5 | 25 | bare soil | Central Namib Desert (Africa) | Feigenwinter *et al.* (2020) |
| ML | 9 | 15.2 | bare soil with biological soil crusts and the grass *Stipa tenecissima* | Balsa Blanca and El Cautivo (Spain) | Uclés *et al.* (2013) |
| ML | 15 and 55 | 25 and 18.6 | soil with biological soil crusts | Wadi Mashash Experimental Farm, Negev desert (Israel) | Ninari and Berliner (2002) |

## 4.4 NRW inputs at CH-FRU

NRW inputs occurred on approximately one third of the nights and were thus a frequent water input. The NRW inputs measured by our ML system represent conservative estimates under certain conditions, because drainage water flow from the ML pots was not measured. Under conditions with water lost via drainage, NRW inputs would be underestimated. Especially during and shortly after intensive rainfall periods, when drainage water flow is more likely (see Appendix F, Fig. F1 and Table F1), the application of the ML system is limited. During transition periods, shortly after rainfall, e.g. during nights when the sky clears after rainfall, NRW inputs may be underestimated. Therefore, we excluded such periods (see Eq. 1) from the analysis and limited our analysis for dry periods. Our longer-term NRW estimates might thus be conservative estimates if rainfall periods are included in the total hydrological input. At our site, drainage water flow from the ML pots reached low levels rather quickly after rainfall events (see the Appendix E and F for more details). Nevertheless, depending on soil characteristics and conditions, drainage water flow could persist for longer time (Fig. F1 and Table F1). Under such conditions, the ML system provides conservative estimates of NRW inputs, because we set NRW input to 0 mm when there is rainfall and/or drainage flow percolating out of the soil monolith. A possible modification of the ML system to also quantify such drainage flow accurately is suggested in the Appendix E with an additional sensor as indicated in Fig. 1b:p. We used three outlets (Fig. 1b:j) to ascertain that drainage is not hindered, but if a sensor to quantify drainage is added, the ML pot should only have one drainage hole with a sensor, from which reliable quantitative estimates of drainage losses can be obtained.

NRW inputs were especially high under conditions when rainfall was absent, e.g. in April, the month with the lowest rainfall. NRW inputs were not influenced by potential NRW input duration, thus there was also a high probability for NRW inputs to occur during summer months, the main growth period of temperate grasses and forbs. In fact, the monthly average NRW inputs were similar to the NRW inputs that were measured in spring and autumn months, when NRW inputs are expected to be highest. This indicates a high ecohydrological relevance of NRW inputs for temperate grassland ecosystems,

especially during hot and dry periods. However, the effects of these frequent NRW inputs on plant
water status have still to be investigated.

Besides studying the effects of NRW inputs on temperate grassland species during hot days with low
soil moisture, a special focus should be directed to the effects of NRW inputs during periods with high
soil moisture, when no soil water stress is present. NRW inputs could be beneficial even under such
conditions, when simultaneously atmospheric demand is high (high energy input, high vapor pressure
deficit). NRW inputs could reduce leaf temperatures by the re-evaporative cooling effect and thereby
reduce water stress during early morning hours and consequently increase productivity (Dawson and
Goldsmith, 2018). However, leaf wetting by NRW inputs could also be disadvantageous during periods
with no soil water stress. Leaves covered by water droplets from NRW inputs could show reduced gas
exchange due to lower gas diffusivity through the water layer. Thus, the development of the ML system
and measuring NRW inputs with high accuracy are crucial steps to address ecohydrological processes,
but further investigations are necessary to understand physiological effects on grasslands.

## 5 Summary and conclusions

The aim of this study was to develop a high accuracy ML system for the quantification of NRW inputs
that overcomes existing drawbacks. The ML system comprised a comparatively large and deep ML pot
in the size class of 25 cm diameter × 25 cm depth in combination with an unprecedented weighing
accuracy. This ML size allowed natural plant growth and such a ML system can therefore be used in
different ecosystems with most short to mid-size statured grasses and forbs or similar vegetation up to
ca. 40 cm. Ancillary sensors allowed differentiating among different types of NRW inputs. Our study
shows that the ML system represents natural conditions very well. The plant height was not
significantly different between ML pots and the control (surrounding). Plant canopy temperatures of
ML pots were close to canopy temperatures of the surrounding during a nocturnal period when NRW
input took place. However, additional continuous canopy temperature measurements in follow-up
studies could allow to more clearly distinguish dew formation from water vapor adsorption and to
identify if canopy temperature drops below dewpoint temperature. If this is not the case, and other
factors like rainfall and fog can be excluded, a weight increase might then be related to water vapor
adsorption. Furthermore, canopy temperature measurements would clarify if a leaf wetness sensor alone
is sufficient to distinguish between dew and water vapor adsorption events. Soil temperatures were
higher in ML pots, especially during the day. This could influence the hydraulic characteristics of soil
water, the heat balance of the soil and in consequence lead to biased latent and sensible heat fluxes.
Thus, further ML studies should primarily focus to get rid of soil temperature differences between ML
pots and the surrounding soil. In addition, the ML system could be further improved by adding water
flow or water droplet sensors at the ML pot outlets to measure drainage water flow (see Appendix E),
with the goal to avoid underestimation of NRW inputs shortly after intensive rainfall events or during
soil conditions when drainage water flow persists for longer time (see Appendix F). With our ML
system, we were able to resolve mass changes on a 15 kg pot with an accuracy of ± 0.25 g, which
corresponds to ± 0.005 mm of water input. This accuracy allows determining typical water gains by

dew, hoar frost, fog, rime or water vapor adsorption on the order of 0.021 to 0.4 mm in a single night. The study revealed that, NRW inputs occurred frequently and provided on average of all NRW events 0.12 mm of water. Such quantitative estimates will be essential to assess the role that NRW inputs might have on temperate grasslands during summer drought conditions. However, longer-term NRW input measurements would allow to see whether the seasonal pattern of NRW inputs are constant over time, or if they are influenced by weather conditions and thus vary from season to season. Moreover, the effects of NRW inputs on plant physiology in grassland ecosystems have still to be elucidated more carefully, to assess the importance of such water inputs during ongoing climate change such as projected prolonged heat periods in the months of main vegetation growth.

## Appendix A: Location map

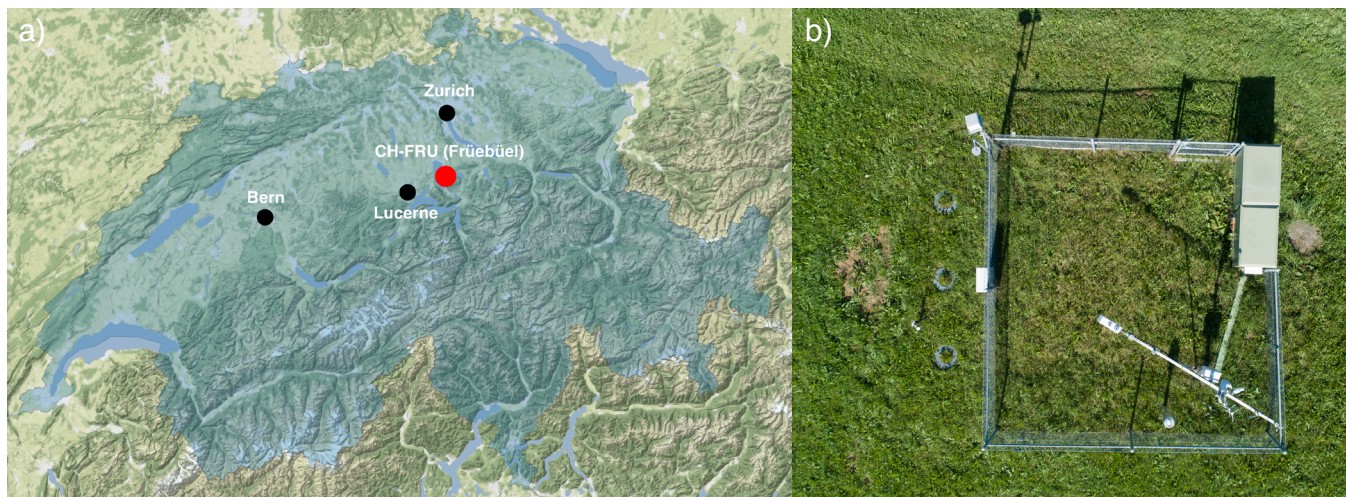

**Fig. A1. a) The red dot indicates the location of the CH-FRU site within the Swiss borders (blue). The black dots indicate the cities of Zurich, Bern and Lucerne. Map tiles by Stamen Design, under CC BY 3.0. Data by OpenStreetMap, under ODbL. b) Aerial photograph taken with a drone of the CH-FRU site. On the left of the fenced area the three ML are visible.**

## Appendix B: Installation procedure and soil monolith preparation

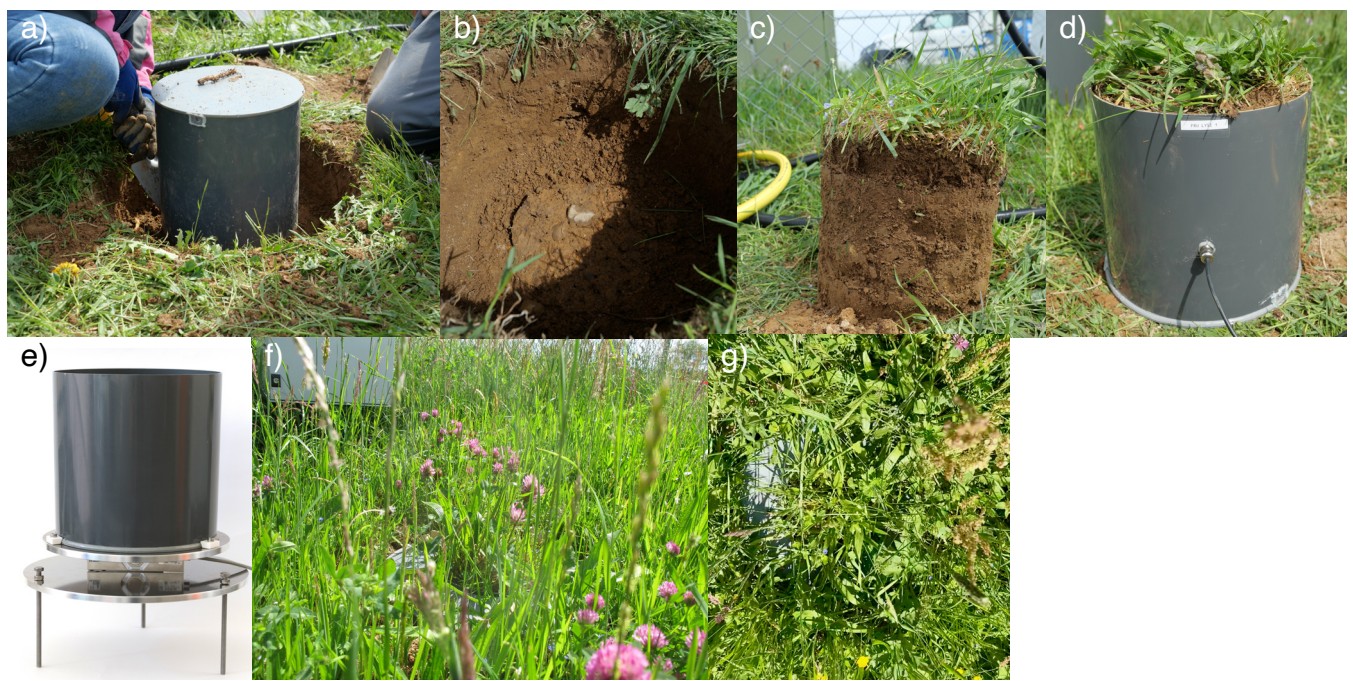

**Fig. B1. Photographs of single ML pots during (a-e) and after installation (f-g) at CH-FRU. a) First step to retrieve an undisturbed soil monolith. An empty ML pot was placed upside down, then the soil around the ML pot was removed with small shovels. Afterwards the ML pot was gently pressed into the soil. b) The contact of the monolith with the soil was cut at the bottom with a spade. C) The monolith was removed from the ML pot and carefully transferred to a second ML pot. d) Monolith ready for installation at the weighing platform. e) Empty ML pot on a weighing platform. The weighing platform is standing on the adjustable support feet. f) Lateral view of an installed ML. g) Top view of an installed ML.**

To retrieve an undisturbed soil monolith with intact grass vegetation, we used an empty ML pot that was placed upside down at the place of interest from where the monolith was to be retrieved. First, we trenched the soil with a long spade around the ML pot. Then we removed the soil around the ML pot with small shovels, which allowed pressing the ML pot into the soil. We continued until the top of the ML pot was at ground level. Finally, the contact with the soil could be cut at the bottom with a spade. The reversed soil monolith was carefully taken out from the ML pot and three people collaborated to transfer it to a second ML pot to be upright again. The ML pot was then ready for installation on the weighing platform. The weighing platform was levelled out by adjusting the three adjustable standing feet with a prolonged hexagon socket wrench. The final position was fixed with the counter nut by using an open-end wrench.

## Appendix C: Data collection, storage, and delivery

Data from all sensors were collected by an Arduino-type MEGA 2560 PRO microcontroller (RobotDyn, Zhuhai, China), which was installed on a custom-made printed circuit board (PCB). The voltage signal

coming from the load cells was digitised by a 24-bit analog-to-digital converter for weigh scales (LM711, SparkFun Electronics, Niwot, USA). For each load cell, a separate analog-to-digital converter was used. After collecting and processing the data of the load cells and the other sensors, the data were
stored as one-minute averages on a micro-SD card (MicroSD 16 Gb, Kingston Technology Company Inc., Fountain Valley, USA) inserted in the slot of a micro-SD breakout board (MicroSD card breakout board 254, Adafruit Industries, New York, USA). Then, the data were transferred to our data server every five minutes by using Internet of Things (IoT) technology. To send the data, a breakout board (RFM9X LoRa Radio, Adafruit Industries, New York, USA) connected to the open TheThingsNetwork
was used. TheThingsNetwork uses a Long Range Wide Area Network (LoRaWAN) protocol. A real-time clock (DS3231 for PI, HiLetgo, Shenzhen, China) was installed on the PCB to obtain exact timestamps.

### Appendix D: Load cell data low-pass filtering

Load cell data are prone to noise. To cancel the noise related to temperature fluctuations, the load cells
used four strain gauges in a Wheatstone bridge configuration. Thus, noise visible in the data mostly originated from electrical noise, fluctuations in wind speed and atmospheric pressure. To minimize this noise, we used a data filtering algorithm on the microcontroller. The microcontroller measured the load cells nominally at 3.3 Hz in combination with the retrieval of measurements from other sensors. The raw load cell data were then stored in an averaging window (ring memory) with a size of 100 values,
where the oldest values were replaced by the newest ones. The upper and lower 15% of these values within the averaging window were discarded, and the remaining values were averaged. From the low-pass filtered signal, one-minute means were stored on the micro-SD card. For data delivery via IoT, these mean values were further averaged over five-minute intervals to comply with the allowed IoT bandwidth for data transfers.

### 760 Appendix E: Drainage water flow of ML pots

The ML pots were designed to avoid stagnation of water that potentially could impede plant growth by creating anaerobic conditions in the rooting zone. For that reason, a passive drainage water flow path allowed drainage of excess water beyond field capacity. However, to further develop this ML system and use it during and shortly after rainfall periods or to improve the measurements during other periods
when the soil cannot hold excessive water, it is recommended to quantify drainage water flow. This is because NRW inputs increase the mass of ML pots, whereas drainage water flow out of the ML pots reduces their mass. Therefore, if drainage water flow during NRW inputs is non-zero, this would lead to an underestimation of the NRW inputs, as long as no additional sensor is added to the ML pots to quantify this drainage flow.

To assess the required specification of such an additional sensor and to quantify how long drainage water flow of the ML system persists, we investigated three consecutive events (see Table E1):

1) A high intensity, high amount and high duration rainfall event (Fig. E1a, event 1);

2) an evapotranspiration event from sunrise until sunset (Fig. E1a, event 2), and

3) a NRW input event (Fig. E1a, event 3).

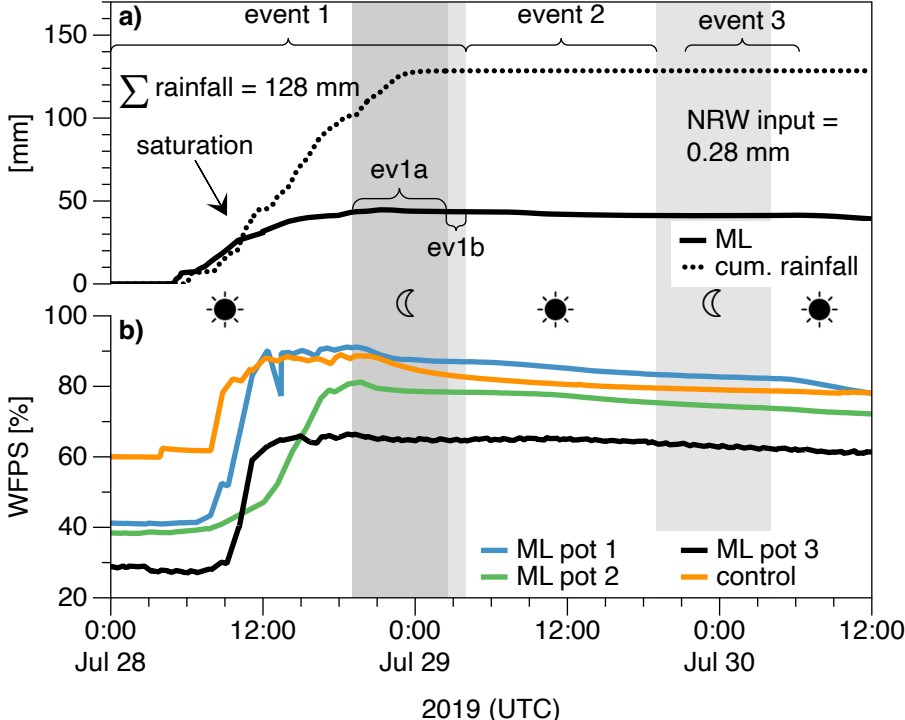


Fig. E1. (a) Cumulative rainfall and ML mass during a rainfall (event 1), an evapotranspiration (event 2) and a NRW event (event 3), from July 28 00:00 until July 30 12:00 UTC. The grey-shaded areas indicate nighttime duration (sunset until sunrise), the unshaded areas indicate daytime (sunrise until sunset). The ML mass and the cumulative rainfall increased with the same rate until the ML pots were almost saturated (indicated with an arrow). Afterwards there was more drainage water lost from the ML

pots than water gained. During the ev1a period (from sunset until the end of rainfall in event1), a rainfall water input of 26.9 mm was observed, but the ML system showed a water gain of only 0.3 mm, the difference between the two measurements corresponds to the (unmeasured) loss via drainage water flow. During the ev1b period (from the end of rainfall until sunrise in event1), there was no rainfall water input, but the ML system showed a water loss of 0.07 mm. During event 2 there was a water loss by evapotranspiration of 2.25 mm. During event 3 (the following night), there was no water loss, but instead a water gain by NRW

input of 0.28 mm. (b) WFPS inside the ML pots and the control, measured at a depth of 15 cm. WFPS reached high values after the rainfall event.

**Table E1. Start, end and duration of the three events used to assess the duration of drainage water flow from ML pots and the specification of a drainage water flow sensor.**

| Event | Start | End | Duration |
|---|---|---|---|
| Event 1 | 28 July 2019 06:03 UTC | 29 July 2019 02:27 UTC | 20 hours and 24 minutes |
| Event 2 | 29 July 04:00 UTC | July 29 19:02 UTC | 15 hours and 2 minutes |
| Event 3 | 29 July 21:18 UTC | July 30 6:17 UTC | 8 hours and 41 minutes |

During event 1, the total amount of rainfall was 128.5 mm. The highest hourly rainfall intensity occurred on 28 July 2019 at 10 UTC with 16.8 mm h$^{-1}$, which classifies as "heavy rain" > 4 mm h$^{-1}$
(Met Office, 2012). ML mass increased as soon as the rainfall event started and increased with the same rate during the rainfall input until ca. 11 UTC. Afterwards the rate of ML mass change, i.e. the slope of the ML mass increase was flattening compared to the cumulative curve of rainfall input: From the beginning of the rainfall event until sunset, the water input was 101.6 mm, whereas the ML system showed an increase of only 36.2 mm. The difference of 65.4 mm most likely corresponds to the losses
from drainage water flow, because of soil saturation during such high intensity rainfall with excessive water being lost. However, WFPS did not reach the 100% mark (Fig. E1b). Note that the 100% WFPS reference was determined from the full year of measurements and is thus relative to spring conditions. Therefore, it is not surprising that this mark was never reached during dry summers, even after heavy precipitation. During such a high rainfall water input, drainage water flow of the ML system was on the
order of 64 % of the rainfall amount. However, water might not only be lost via drainage water flow, but also by evapotranspiration during daytime. To quantify solely drainage water loss, the nighttime period (when no evapotranspiration is expected) was further investigated. We separated the nighttime period in period ev1a, when rainfall occurred, and period ev1b, when no rainfall occurred (Fig. E1a, gray shaded periods).

During the ev1a period (Fig. E1a, period ev1a), from sunset until the end of the rainfall event, the water input was 26.9 mm, whereas the ML system showed only an increase of 0.3 mm. The difference of 26.6 mm (98 %) might be caused by losses from drainage water flow. The water loss rate was 3.6 mm h$^{-1}$. The 34 % higher drainage water loss compared to the daytime period might be due to the lower water holding capacity of the more saturated soil. During the ev1b period, starting after the ev1a period until
sunrise (Fig. E1a, period ev1b), no further water gains and losses were expected, because evapotranspiration was absent during nocturnal conditions with low average wind speed (< 0.6 m s$^{-1}$). During period ev1b, the ML system showed a water loss of 0.07 mm, which corresponds to an average water loss of 0.05 mm h$^{-1}$. This water loss can clearly be attributed to drainage water flow. The rate of drainage water loss was however strongly reduced (by 98%) compared to the ev1a period. Thus,
drainage water flow of the ML system reached very low values within only 1 hour and 33 minutes after this extraordinary high rainfall, showing that even the current ML system can handle high drainage water flows well.

During event 2 with no rain but evapotranspiration, the ML system indicated a water loss of 2.25 mm, which corresponds to an average evapotranspiration rate of 0.15 mm h$^{-1}$. Potentially a drainage water loss could have occurred in the morning hours on July 29. However, the drainage water loss most likely was < 0.05 mm h$^{-1}$, similar to the drainage water flow rate during the ev1b period, just before event 2, shortly after the rainfall event. Since no new rain fell, we expect the drainage water flow rate to decrease with time. In fact, one hour before sunset, a further reduced ML mass loss of only 0.005 mm h$^{-1}$ was recorded. This very low water loss can be either attributed to drainage water loss, or to evapotranspiration as it occurred during daytime. We conclude that the drainage water loss could at maximum be 0.005 mm h$^{-1}$, but was most likely lower due to concurrent evapotranspiration. Thus, the ML system readings were no longer significantly affected by potential drainage water flow after only 15 hours after rainfall.

During event 3, a very large dew event of 0.28 mm occurred, which was above the 95$^{th}$ percentile of all NRW events during the 12 months period considered in this study. Such a large dew event is unlikely to be recorded under conditions when at the same time also a large drainage water flow would have occurred. If this would have happened, the dew water input should have been lower. Thus, it is very unlikely that drainage water flow still occurred during that dew event.

Overall, these three events showed that drainage flow occurred under rainfall conditions and shortly after rainfall events. The current ML system handled large drainage flows well and effectively, i.e. water drained fast, avoiding long-lasting "memory" effects. Drainage flow was lower than 0.005 mm h$^{-1}$ one hour before sunset during event 2, only 15 hours after the last rainfall. However, at other sites with different soil characteristics different drainage flow patterns might occur (See Appendix F) and our ML system might therefore provide conservative NRW inputs and accentuated evapotranspiration rates. If the current ML system were to be used for high rainfall conditions, potential drainage water flow need to be quantified using additional sensors. Without such additional sensors, NRW inputs could be underestimated if the NRW input occurs shortly after a rainfall event and drainage water flow indeed occurs. Consequently, the current ML system is expected to give conservative estimates of NRW inputs, especially if NRW inputs happen directly after a rainfall event.

Potential approaches to quantify the small amounts of drainage flow from a ML system are by installing a water flow sensor or a drip counter at the ML pot drainage water outlets. The maximum rainfall intensity reported above was 16.8 mm h$^{-1}$. With a ML pot diameter of 25 cm (see Section 2.2.1 of the main text), and the extreme assumption that 100% of precipitation contributes to drainage water flow, such an addition must be able to process 13.7 ml min$^{-1}$. If the maximum drainage water flow is however only expected to be <15% of precipitation, then a sensor capable of measuring up to 2000 µl min$^{-1}$ would be an adequate choice.

We recommend using a water flow sensor or a drip counter. One option is a liquid flow sensor (SLF3S-0600F, Sensirion AG, Staefa, Switzerland) that is capable to detect low flow rates of up to ±2000 µl min$^{-1}$. A drip counter can be constructed with two gold electrodes attached to the ML pots drainage water holes with a small gap. If a water droplet passes the gap, an electric circuit is closed which can be counted as a water drop by a datalogger (Meter Group AG, 2020). Calibration of a drip counter is

recommended for accurate measurements of drainage water amount. Sensors measuring drainage water flow would allow to correct for drainage water outflow and would thereby increase the usability of the current ML system for times during and shortly after rainfall events.

## 865 Appendix F: Duration of drainage water flow after heavy rainfall (saturated soils)

Drainage water flow was not quantified in the application of the ML system described here, because the goal was to quantify NRW inputs during dry conditions without saturated soils. To estimate the duration of drainage water flow from the bottom of the ML pot, we used the approach by Zhan et al. (2016) with modifications following Freeze and Cherry (1979) and model input parameters from Rawls et al. (1991)
listed in Table F1. The full equation set used here is provided in what follows.

The relation between the unsaturated hydraulic conductivity $k$, the volumic water content $\theta$ and the pore-water pressure head $\psi$ can be described by the following formula:

$$\frac{\partial}{\partial z_*}\left(k\frac{\partial\psi}{\partial z_*}\right) + \cos(\gamma)\frac{\partial k}{\partial z_*} = \frac{\partial\theta}{\partial t}. \tag{F1}$$

Note that we are only considering the case where $\psi < \psi_{ae}$. In the case where $\psi \geq \psi_{ae}$ both variables
$k, \theta$ are constant.

In order to solve this equation we can substitute $k = k_s e^{\alpha(\psi+\psi_{ae})}$ and

$$\theta = \theta_r + (\theta_s - \theta_r)e^{\alpha(\psi+\psi_{ae})} \tag{F2}$$

and use the product rule

$$\frac{\partial}{\partial z_*}\left(k\frac{\partial\psi}{\partial z_*}\right) = \frac{\partial}{\partial z_*}\left(k_s e^{\alpha(\psi+\psi_{ae})}\frac{\partial\psi}{\partial z_*}\right) = k_s\left(\frac{\partial e^{\alpha(\psi+\psi_{ae})}}{\partial z_*}\frac{\partial\psi}{\partial z_*} + e^{\alpha(\psi+\psi_{ae})}\frac{\partial^2\psi}{\partial z_*^2}\right) = \left(\alpha\frac{\partial\psi}{\partial z_*}\frac{\partial\psi}{\partial z_*} + \frac{\partial^2\psi}{\partial z_*^2}\right)k_s e^{\alpha(\psi+\psi_{ae})}$$
$$\tag{F3}$$

to get

$$\frac{\partial\psi}{\partial t} = \frac{k_s}{\alpha(\theta_s-\theta_r)}\cdot\left[\alpha\left(\frac{\partial\psi}{\partial z_*}\right)^2 + \frac{\partial^2\psi}{\partial z_*^2} + \alpha\cos(\gamma)\frac{\partial\psi}{\partial z_*}\right] \tag{F4}$$

To solve this numerically we assume a uniform saturated ground at $t = 0$ given by $\psi(z_*, 0) = -\psi_{ae}$ and $\frac{\partial\psi}{\partial z_*} = 0$ for all $z_*$.

Moreover, we impose the boundary conditions $\frac{\partial\psi}{\partial z_*} = 0$ at $z_* = 0$ and $\frac{\partial\psi}{\partial z_*} = \cos(\gamma)q$ at the top $z_* = H_*$ where $q$ is the rainfall intensity. Here we choose $q = 0$ to look at the situation after long rainfall events.

We simulated the drying of the 25 cm deep soil monolith using a finite difference approach with $\Delta z_* = 0.01$ m and $\Delta t = 10$ minutes. The procedure carried out at each timestep was: (1) to compute the drainage water loss across the bottom of the soil monolith $P_{er}$ using Zhan et al.'s (2016) equation,

$$P_{er} = k'k_s\cos\gamma, \tag{F5}$$

where $k'$ is the dimensionless ratio of the unsaturated hydraulic conductivity normalized by its value at saturation ($k_s$), $k' = k/k_s$. and $\gamma$ is the slope angle. For the simulations we assumed $\gamma = 0°$; thus, in case of a sloping surface the drying of the soil monolith takes less time ($t_d$) than what we present in Figure F1.

(2) This amount of water was removed from the lowest soil layer $\theta(z_*=0)$. (3) Then the updated soil water content profile $\theta(z_*)$ was converted to an updated pressure head profile $\Psi(z_*)$ using the relationship in Eq. (F2) solved for $\Psi$,

$$\Psi = \frac{1}{\alpha}\ln\frac{\theta-\theta_r}{\theta_s-\theta_r-\Psi_{ae}}, \tag{F6}$$

(4) Then the drainage flow rate for all soil layers was computed with Eq. (F5), and the respective amount was transferred from each layer to its lower adjacent layer. (5) Then the $\theta(z_*)$ profile was converted to $\Psi(z_*)$ and the change over time from Eq. (F3) was added, and then the $\theta(z_*)$ profile was updated accordingly before the next timestep was simulated.

The threshold for the end of the drainage period was set to one drop of water per day percolating out of the soil monolith's bottom (0.05 mm d$^{-1}$, or 0.35 µm at the $\Delta t = 10$ minutes timestep).

Following Timlin et al. (2004) we used the Brooks-Corey pore size distribution $\lambda$ tabulated in Table F1 in combination with the effective porosity $\phi_e$ (m$^3$ m$^{-3}$) defined as the difference between total porosity $\phi$ (m$^3$ m$^{-3}$) minus the water retained in the soil matrix at a suction pressure of

$-33$ kPa ($\theta_{33}$; m$^3$ m$^{-3}$), $\phi_e = \phi - \theta_{33}$,

$$k_s = 0.000259 \cdot 10^{0.6\lambda} \cdot \phi_e^{2.54}. \tag{F7}$$

The results for different soil textures are shown in Figure F1. Given the initial condition that the soil monolith is completely water saturated at $t = 0$ our results show rather conservative estimates how long water is percolating out of the ML pot after heavy rainfall or long rainfall that saturated the entire soil volume (which typically takes a few days to a week with precipitation).

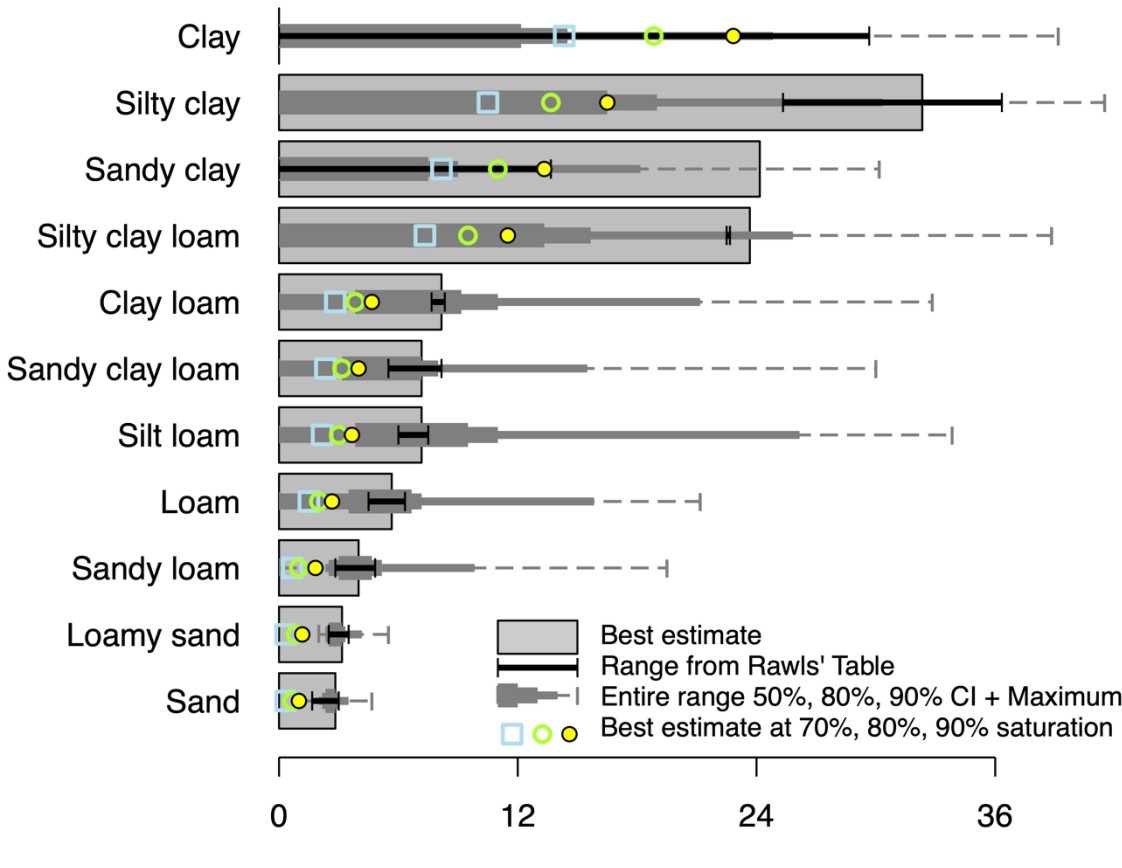

Fig. F1. Estimated duration of percolation ($t_d$) at the bottom of a 25-cm soil monolith in a ML for various soil types. Bars show the best estimate for each soil type for completely saturated soils, and whiskers show the range that results when the parameter range given by Rawls et al. (1991) in Table F1 are used. Symbols show the reduced $t_d$ when the average water content of the soil monolith is 90% of its saturation (yellow circles), 80% (green circles), or 70% (blueish circles). Because the system is highly nonlinear, the parameters given in Table F1 are not resulting in the full range of $t_d$, hence we added the maximum that can be obtained with intermediate model parameters for each soil type (dashed whiskers) and the 70%, 80% and 90% two-sided confidence interval (gray bars of varying width, see legend) for all $t_d$ resulting by combination of parameter values within the bandwidth given in Table F1.

Most soils on average fall dry within less than 24 hours; the absolute maximum was modeled for silty clay which can produce drainage up to 41.5 hours. At the sandy end short maximum $t_d$ are realistic because of easy drainage of soils with high sand content, whereas the results at the clay side show a range from no drainage up to 30.0–41.5 hours can be explained by the high capillary retention of water that retains more water inside the soil volume without generating drainage water flow. The modeling however is based on a traditional micropore flow approach, whereas macropore flow (e.g. Alaoui and Eugster, 2004) is not explicitly represented in the model. But the range of parameter estimates in Table

F1 seems to include also macropore flow via parameter combinations that result in $t_d = 0$ hours, which is most likely not realistic, but should be interpreted that in the presence of macropore flow (wormholes, dry cracks in clay) the drainage is restricted to very short intervals even after soils were fully saturated). Thus, in reality most but not all soils will most likely not produce measurable drainage after one day or so. Adding a sensor to measure drainage water flux (item q in Fig. 1b) is recommended if in contrast to
this study the entire hydrological soil water budget shall be quantified, and not only the NRW gain during dry and drought periods.

| Soil Texture | Total Porosity | Residual Water Content | Air Entry Pressure | Pore Size Index λ | Water Retained at –33 kPa | Hydraulic Conductivity | Drainage Time |
|---|---|---|---|---|---|---|---|
| Class | $\theta_S$ | $\theta_R$ | $\Psi_{EA}$[a] | $\lambda$ | $\theta_{33}$[a] | $k_s$ | $t_d$ |
|  | $m^3\ m^{-3}$ | $m^3\ m^{-3}$ | m | – | $m^3\ m^{-3}$ | $m^3\ s^{-1}$ | h |
| Sand | 0.437 | 0.020 | 0.0726 | 0.592 | 0.091 | 3.961E-05 | 2.8 |
|  | [0.374, 0.500] | [0.001, 0.039] | [0.0136, 0.3874] | [0.334, 1.015] | [0.018, 0.164] | [2.981E-05, 6.595E-05] | [2.2, 3.0] |
| Loamy sand | 0.437 | 0.035 | 0.0869 | 0.474 | 0.125 | 2.587E-05 | 3.0 |
|  | [0.368, 0.506] | [0.003, 0.067] | [0.0180, 0.4185] | [0.271, 0.827] | [0.060, 0.190] | [1.892E-05, 4.352E-05] | [2.5, 3.5] |
| Sandy loam | 0.453 | 0.041 | 0.1466 | 0.322 | 0.207 | 1.147E-05 | 4.0 |
|  | [0.351, 0.555] | [-0.024, 0.106] | [0.0345, 0.6224] | [0.186, 0.558] | [0.126, 0.288] | [7.576E-06, 1.956E-05] | [2.8, 4.7] |
| Loam | 0.463 | 0.027 | 0.1115 | 0.220 | 0.270 | 5.378E-06 | 5.7 |
|  | [0.375, 0.551] | [-0.020, 0.074] | [0.0163, 0.7640] | [0.137, 0.355] | [0.195, 0.345] | [4.017E-06, 7.648E-06] | [4.5, 6.2] |
| **Silt loam** | **0.501** | **0.015** | **0.2076** | **0.211** | **0.330** | **3.906E-06** | **6.8** |
|  | **[0.420, 0.582]** | **[-0.028, 0.058]** | **[0.0358, 1.2040]** | **[0.136, 0.326]** | **[0.258, 0.402]** | **[3.070E-06, 5.215E-06]** | **[5.8, 7.5]** |
| Sandy clay loam | 0.398 | 0.068 | 0.2808 | 0.250 | 0.255 | 2.617E-06 | 7.0 |
|  | [0.332, 0.464] | [-0.001, 0.137] | [0.0557, 1.4150] | [0.125, 0.502] | [0.186, 0.324] | [2.321E-06, 3.513E-06] | [5.3, 7.5] |
| Clay loam | 0.464 | 0.075[b] | 0.2589 | 0.194 | 0.318 | 2.554E-06 | 7.7 |
|  | [0.409, 0.519] | [-0.024, 0.174] | [0.0580, 1.1570] | [0.100, 0.377] | [0.250, 0.386] | [2.785E-06, 2.595E-06] | [7.0, 8.0] |
| Silty clay loam | 0.471 | 0.040 | 0.3256 | 0.151 | 0.366 | 1.042E-06 | 22.5 |
|  | [0.418, 0.524] | [-0.038, 0.118] | [0.0668, 1.5870] | [0.090, 0.253] | [0.304, 0.428] | [1.180E-06, 9.551E-07] | [21.8, 21.2] |
| Sandy clay | 0.430 | 0.109 | 0.2917 | 0.168 | 0.339 | 7.414E-07 | 23.2 |
|  | [0.370, 0.490] | [0.013, 0.205] | [0.0496, 1.7160] | [0.078, 0.364] | [0.245, 0.433] | [1.466E-06, 2.962E-07] | [0.0, 11.7] |
| Silty clay | 0.479 | 0.056 | 0.3419 | 0.127 | 0.387 | 7.203E-07 | 31.0 |
|  | [0.425, 0.533] | [-0.024, 0.136] | [0.0704, 1.6620] | [0.074, 0.219] | [0.332, 0.442] | [6.881E-07, 7.955E-07] | [24.5, 34.5] |
| Clay | 0.475 | 0.090 | 0.3730 | 0.131 | 0.396 | 4.919E-07 | 0.0 |
|  | [0.427, 0.523] | [-0.015, 0.195] | [0.0743, 1.8720] | [0.068, 0.253] | [0.326, 0.466] | [8.415E-07, 2.541E-07] | [0.0, 28.5] |

[a] geometric mean values were used from Rawls et al.'s (1991) table
[b] this value was considered a typographic error in Rawls et al.'s (1991) table and corrected here (×0.1)

**Table F1: Model parameters used in estimating duration time ($t_d$) until less than one water droplet per square-meter and day (0.05 mm d$^{-1}$) is draining out of a 25 cm deep soil monolith volume in a ML. Hydraulic conductivity ($k_s$) was computed with Eq. (F7). The best estimate for each parameter is complemented by a range suggested by Rawls et al. (1991) shown within brackets. The parameters for the silt loam soil at the field site is highlighted in boldface for reference.**

## Appendix G: NRW inputs vs. night time duration

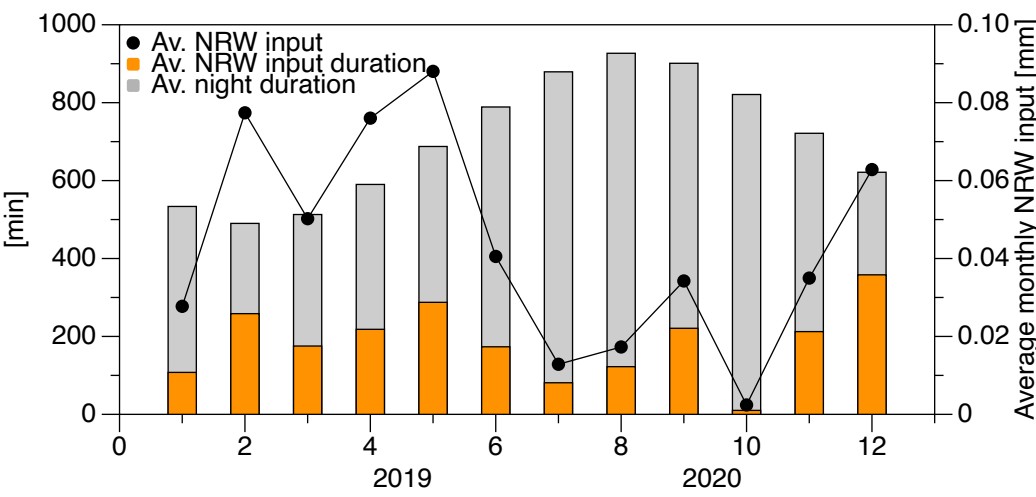

**Fig. G1. Average monthly NRW input with average monthly NRW input duration and average night duration (potential NRW input duration).**

**Data availability.** Data are stored at the ETH Zurich research collection at: https://doi.org/10.3929/ethz-b-000488747 (Riedl, 2021).


**Author contribution.** AR and WE designed the ML system. AR built the ML system and installed it together with YL. AR carried out maintenance, experiments, data collection and data analysis. JE and WE added Appendix F. NB, WE commented on the results of data analysis. AR wrote and revised the manuscript, with contributions and feedbacks by YL, NB and WE.


Competing interests. The authors declare that they have no conflict of interest.

Acknowledgements. We are grateful to Paul Linwood for his excellent work with electronics and installation assistance in the field, and we thank Markus Staudinger, Philip Meier and Thomas Baur for
their technical support. We also thank Patrick Flütsch for constructing the ML pots and lids.

Financial support. This research was founded by the Swiss National Science Foundation (grant no. 175733).

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
