# Peer review of "Technical note: High accuracy weighing micro-lysimeter system for long-term measurements of non-rainfall water inputs to grasslands"

_Hydrology and Earth System Sciences, 2021_

## Author Comment (AC1)

Anonymous Referee 1:

As an important component of the ecohydrological cycle, it is always difficult to analyze and quantitatively characterize Non-rainfall water (NRW). In this paper, the author designed, manufactured and tested a novel micro-lysimeter (ML) system, which has high precision and weighing range, and overcomes some defects of the existing lysimeters. Different types of NRW inputs, such as dew, hoar frost, fog, rime and the combinations among these, are well distinguished by auxiliary sensors. At the same time, it is also similar to the surrounding environment in terms of canopy and soil temperature, plant growth and soil humidity. The author applied the ML system at a field site in Switzerland. Through the monitoring of a hydrological year, different NRW events were effectively distinguished and the NRW inputs was quantified. In general, the paper has good innovation and practical value, but some parts need to be improved, and major revision are suggested. The specific opinions are as follows.

1. It is recommended to attach the location map of the study site in Switzerland and the real photos of the ML system;
2. How deep is the groundwater level? Should the impacts of groundwater be considered?
3. Line 215-220, it may be difficult to transfer the soil body from one ML to the second. How to ensure the stability of soil during transfer?
4. Do you think freezing-thawing would have impacts? Should the accuracy of the ML change at different temperatures?
5. Can the ML distinguish the influences from dust or other drifting materials falling on it?
6. According to the author's statement, the ML can only distinguish windless conditions, so it cannot work on windy days? NRW input over a year in Figure 8 might be underestimated?
7. The ML can quickly discharge the excess water after precipitation, but this would also cut off the evapotranspiration channel, which might misrepresent the conditions in the nearby soils. Will it affect NRW?
8. In figure 7 (b), it is described as WFPS in the text, but not in this figure.
9. Different types of NRW inputs in Figure 8 can be represented in different colors. The same is for Figure 9.

Response to Referee's comments:

We thank the anonymous reviewer for his/her constructive feedback and inspiring questions. We will do our best to improve our manuscript accordingly. Our response to reviewer's suggestions and critique is given in blue indented text point by point.

1. We will add a location map and real photographs of the ML system to the Appendix.

2. The Früebüel site (CH-FRU) is in a local groundwater zone (see GIS map below), but without  groundwater level depth information. The reason for this is that the CH-FRU site is located on a hill (red circle in map below) on a glacial moraine with undulating bedrock below, hence the depth to groundwater should be deepest where our long-term site is placed. All our sites (not only the one shown here) are either in mountain terrain without large-scale alluvial deposits where a homogeneous aquifer could develop, and in the one case where our file site (CH-CHA, not included in this analysis) is in a valley with a well-defined aquifer, the water table of this aquifer is ca. 10 m below surface. So the topographic setting at CH-FRU does not allow for stagnant water leading to high water tables, but it can lead to wet soils when intensive precipitation is observed over longer periods, which however can drain in all directions. The disconnection of the ML pot to the underlying soil and in consequence to the groundwater body is however a general lysimeter problem. We will add a statement to the manuscript.

[Figure]

Extent of the groundwater zone around the Früebüel site (CH-FRU), taken from the cantonal GIS (dimension of screenshot is 1700 m × 960 m, taken from maps.zg.ch – only available in the official language which is German). The isolines around the site (red circle) show the drainage to be in all directions.

3. Indeed! This is quite a challenge. From our experience the monolith breaks most likely at the bottom, whereas breaking at the edges is not a big issue. At the time of installation, we were three people at the site and could manage to retrieve three undisturbed soil monoliths after one failure. We found that this work is best done when the soil is not completely dry or extremely wet to avoid any breaking of the monolith. We will add more details about this process, furthermore we will add some photographs to describe the process better.

4. Freezing-thawing could damage PVC parts of the ML, however this did not occur during the study period. We will add that it could potentially occur.
The used load cells are temperature compensated. Four strain gauges in a Wheatstone bridge configuration are part of each load cell. The changes in resistance of the strain gauges that would occur solely due to temperature changes are thus balanced, because

all four strain gauges (two positive, two negative) behave in the same way with temperature change and thus effectively cancel the temperature change signal. The remaining, minute residual error is corrected by special Nickel elements that are connected to the Wheatstone bridge. We will add to the methods section that the load cell was temperature compensated.

5. Thank you for pointing this out. Materials falling on the ML pots would lead to a sudden mass increase, which can be identified in the data. However, slow deposition of dust/sand particles could be misinterpreted as NRW inputs. We don't think that this is a big issue at the CH-FRU site (high vegetation cover), but it could be a big issue in deserts and arid regions, where the soil vegetation cover is low. We will add that depositing materials that accumulate on the ML over time like sand could be an issue.

6. The ML system was designed to measure NRW inputs, which occur during nights with low wind speeds. The probability for dew formation decreases below 5% when wind speeds are smaller than 0.4 m s$^{-1}$ or bigger than 1.9 m s$^{-1}$ (Zhang et al., 2014). Thus, wind might not be a big bias source for NRW quantification, because wind speeds are low during NRW inputs. Higher wind speeds could exert a force on the ML pot and increase thereby the mass, which would lead to an overestimation, not to an underestimation of mass. High nightly wind speeds are the main driver of night time evapotranspiration (antagonist of NRW inputs) instead of dew formation (Groh et al., 2019; Padrón et al., 2020). We will add this information for clarification.

7. The water source of NRW inputs is atmospheric water vapor. The atmospheric water vapor can also stem from evaporated soil water, termed distillation. Distillation can contribute up to 42 % of the condensed water on foliage during dew nights (Li et al., 2020). Thus, a ML pot that is cut off from the evapotranspiration channel might contribute less to the distillation process. However, in the boundary layer atmospheric water vapor is distributed via turbulence and thus the ML is subject to the same atmospheric water vapor as the surrounding field. This cut-off might still lead to the same amount of condensation (e.g. dew formation on plants). However, prolonged altering of soil moisture could have an influence on plant growth and in consequence to NRW inputs (please have also a look at our response to a similar comment by reviewer 2).

8. Thanks for pointing this out. We used once "WFPS" instead of "Soil temperature" which created some confusion. We will correct it.

9. We will change it according to your suggestions.

Groh, J., Pütz, T., Gerke, H. H., Vanderborght, J. and Vereecken, H.: Quantification and Prediction of Nighttime Evapotranspiration for Two Distinct Grassland Ecosystems, Water Resour. Res., 55(4), 2961–2975, doi:10.1029/2018WR024072, 2019.

Li, Y., Aemisegger, F., Riedl, A., Buchmann, N. and Eugster, W.: The role of dew and radiation fog inputs in the local water cycling of a temperate grassland in Central Europe, Hydrol. Earth Syst. Sci. Discuss., 1–27, doi:10.5194/hess-2020-493, 2020.

Padrón, R. S., Gudmundsson, L., Michel, D. and Seneviratne, S. I.: Terrestrial water loss at night: global relevance from observations and climate models, Hydrol. Earth Syst. Sci., 24(2), 793–807, doi:10.5194/hess-24-793-2020, 2020.

---

## Author Comment (AC2)

Anonymous Referee 2:

We thank the anonymous reviewer for the constructive and detailed feedback. We will do our best to improve our manuscript accordingly. Our response to reviewer's critique is given in blue indented text after each point addressed by the reviewer.

The paper introduces a microlysimeter for the specific purpose of measuring the deposition of water on the surfaces of the soil and the vegetation by means other than rain (dew, fog, etc.). The deposited amounts of water are quite small, yet the paper argues that they could be important for the vegetation during dry periods. The design of the set-up, with its high observation frequency and high-resolution mass measurements is explained in some detail. The instrument has been in operation in the field. Its performance is reported and evaluated.

General comments

Overall, the paper gets across the relevance of MLs and the improvements the authors have made to earlier designs. The material fits the HESS mission and its readership.

That being said, the paper is wordy and tedious at times. The authors go in so much detail that is hard to follow the line of thought. In contrast, Figure 1 leaves out many technical details, and photographs of the ML and its components are not given.

> The schematic figure 1 leaves out only 4 connection bolts, adding these connection bolts would make the figure busier. But we agree that Fig. 1 is a schematic and not a technical drawing with all details. If the latter is required, we could include a technical drawing in the Appendix, if the Editor decides this should be added. In Figure 1 we will rearrange some numbers to clearer indicate which part is meant and add photographs of the ML to present it in a clearer form.

The Introduction is comprehensive but incoherent, and would benefit from a careful revision. It can be shortened a little perhaps.

> Thanks for this suggestion. We will shorten and rearrange the introduction to increase coherence.

In Materials and Methods, there is no information at all given about the soil of the experimental site. The technical details of the ML setup and its installation and operating procedures need to be described better. The rest of the Methods are very detailed, sometimes about well-established techniques. You can shorten the text there.

> We will add information about the soil at the site. We think that adding photographs of the ML system and photographs of the installation process will also help to better understand installation and operating procedures. Moreover, we will shorten the text in the Methods section where possible.

The authors make a point about separating the various NRW modes, but given the small total flux, I do not understand why this is so important. On the other hand, the differences in soil temperature between the ML and the surrounding soil is downplayed, even though its various effects may linger into the night, when NRW occurs.

> Separating various NRW inputs might be less important for plant biologists, which focus more on the reactions of the plants to water supply and are less focused on where this water came from. However, e.g. for meteorologists and agrometeorologists it is crucial how often, when and under which meteorological conditions NRW input occurs. E.g. the formation and dissipation of fog is a process that is not clearly understood. We think adding the separation of various NRW inputs makes this manuscript more interesting for a broader readership, but we agree that ecohydrologists may be mostly interested in the total NRW inputs. For the difference in soil temperature, please have a look at the response on the comment to Section 5.

The material on which the paper is based is solid, but the presentation is not so good. The paper would benefit from a thorough rewrite that increases its coherence and clarity, and reduces its size a little.

The paper mentions a supplement but I could not find that, other than the data set.

> We appologize for this error, the supplement has been included as an appendix, but we overlooked the need for a change in wording. We will change "supplement" to "appendix".

Detailed comments

L34: The sentence introducing the new ML looks out of place here. You are still developing the argument for its necessity, only moving from general terms to a specific ecosystem.

> We agree and will rearrange the introduction.

L39: If rainfall ('RW') is absent, NRW necessarily IS the only atmospheric source of water, because RW and NRW are mutually exclusive and complementary.

> We will change the sentence to: "During drought periods, NRW inputs are the only available atmospheric water source."

L43: 'another temperate site' You did not mention the first one.

> We will delete it.

L51: What do you mean by plant water status?

> We mean "plant water potential", the term is quite well established in the plant physiological literature (see e.g. Jones, 2006, doi:10.1093/jxb/erl118 for a definition), but we realize now that this is not the case in the field of ecohydrology. We thus will clarify this term accordingly.

L56 and 320: You mention dew deposition on soil, but earlier you stated that does rarely occur.

> It does rarely occur, but still there is the possibility that it occurs.

You need to have a careful look at the Introduction. Although I agree with the arguments it presents, they are presented in a confusing order. Above I mentioned the Introduction of a lysimeter in the middle of a discussion about NRW. Elsewhere too you jump between general statements and location-specific arguments without a logical connection between the two. This compromises the coherence of the text and disrupts the flow of thought. All the elements the Introduction needs are there, but please present them in a more coherent order and use more paragraphs as compartments for different focal points of the Introduction.

> We will rearrange the introduction as recommended by the reviewer and will delete and shorten where necessary and meaningful.

Section 2: Some subsections are not directly important to understand the ML setup. Perhaps the very detailed material can be placed in the supplement.

> We will move these less directly important subsections to the appendix, i.e. 2.2.3 Soil monolith preparation.

Section 2.1: Please give some information about the soil.

> We will add: "The soil at the site is a silt loam mixture (56% silt, 37% sand, 7% clay), with a bulk density of $1.12 \pm 0.03$ g cm$^{-3}$ and an organic C content of $4.4 \pm 0.2\%$ (Stiehl-Braun et al., 2011)."

L124: What is FluxNet? Do we need to know?

> This is part of the site information to tell readers in which network this site is embedded. Below is a statistics extracted from the Scopus database on the increasing importance of FluxNet in the field of eddy covariance flux measurements of $H_2O$ and $CO_2$ fluxes. It is thus not surprising that the name is new to some readers, but this might change rapidly if the spread of FluxNet data usage continues at its current pace. Thus, we prefer to keep this information and expect that readers not interested in it will simply ignore.

[Figure]

L154: Instead of the width and the thickness it is probably better to give the outer and inner diameter of the ML tube.

We will change it according to your suggestions.

L173-174: How do you level the load cell and the ML during installation in the field? I cannot see the adjustment screws in the figure or how you can reach them during the installation process.

The adjustment screws (=adjustable support feet) are shown in Fig. 1 as component "h". We will add a further component to Fig. 1 (counter nut) to explain the "H" like structure better. Furthermore, we will add a photograph of the weighing platform and the adjustable support feet, which are basically machine screws (bolts) with a counter nut (otherwise the weighing platform would move). We could reach them with a prolonged hexagon socket wrench. We will add that to the text.

Fig. 1: What are the structures on either side of the load cell that look like an H on its side? The text mentions machine screws (bolts?) but I cannot find these in the figure. I also think it would be helpful to add a photograph of the instrument to the figure, as well as a scale or a set of dimensions within the drawing.

The drawing has no scale, because it is a schematic drawing. Dimensions are mentioned in the text. But adding detailed photographs to our revised manuscript version will indeed improve the understanding by the readers.

Section 2.2.3: I estimate a filled ML weighs about 100 kg. How did you handle it when you excavated the monolith and when you installed the monolith in its field setup?

The ML mass was lower than 20 kg for all three ML (ML pot size of 25 cm diameter x 25 cm depth). The maximum capacity of the load cell was also 20 kg. At the day of installation, we were three people at the site, and we were able to transfer the monoliths to the ML pots and afterwards the whole weighing platform to the outer part. For heavier weights one would need a crane, we agree, but with this weight it is not an issue.

The ML pots were closed on one side. In order to transfer the monolith from the sampling pot to its ML pot you need to place both pots with the open sides against each other, so you end up with a cylinder that is closed on both sides. How did you transfer the monolith from one half of that cylinder to the other?

We will add some photographs to the appendix, we think that helps to understand this process better. Basically, we took it out from one ML pot and transferred it by hand, in upright position, to another ML pot.

L240: What was the size of the averaging window?

We will add the size of the averaging window was 100 values (this was the maximum that this microcontroller could handle). Thus, with a sampling frequency of 3.3 Hz, the averaging window had a length of 30.3 seconds.

Section 2.2.6: it is helpful to mention here somewhere what the resolution of the load cell is converted to mm water layer.

> This information is not available. The load cell itself outputs only a voltage, that's why also the company does not provide such a measure. The resolution depends on the parts of the system, e.g. if one uses a 8-Bit or a 16-Bit analog digital converter, the calibration range and many other factors. We provide information about the resolution of our system in Section 3.1, thus in the results section, not in the methods section, because this is a setup-dependent value. Here we specify the SE of the measurements of the three ML as ± 0.31, ± 0.14 and ± 0.11 g, respectively. We'll add the lacking units to all reported figures in the revisions.

L279: Why the < sign in the sensitivity of the temperature sensor? Now we still do not know its sensitivity.

> The company (Testo AG, Lenzkirch, Germany) provides the sensitivity for this instrument in this way: "Sensitivity: < 50 mK at +30 °C". The sensitivity might be temperature dependent, so it could be in the range from 0 to 50 mK, depending on temperature. Thus, reporting the value with a < sign means "better than" or "at most" and is thus a conservative estimate of the true sensitivity.

L283-284: ,we considered standard deviation to account for spatial variability.' I do not understand this.

> We will change it to: "To compare thermal images of the ML surface with the control, we compared the variance (F-test). Data were bootstrapped to reduce sample size from > 30k to 30 samples using the scikit-learn package in Python (Pedregosa et al., 2011)." Furthermore we will add to the results section: "The variance of canopy temperature between the ML vegetation and the control were not statistically significant different (F-test, p > 0.05, n = 30)."

L327: Accuracy of what? In the section that follows you use the term accuracy a lot, but I believe you sometimes mean precision (e.g., https://www.mccdaq.com/TechTips/TechTip-1.aspx).

> We will change to: "Accuracy of the ML system". In the text we will carefully check the correct usage of accuracy vs. precision. Where a calibration with calibration mass was possible, we will retain the term accuracy. In cases where an absolute standard could not be used, we will adopt the term "precision" instead.

L328: Please give more significant digits for the correlation coefficient.

> We will give more significant digits for the correlation coefficient.

L365: with your measurement frequency, individual eddies in the near-surface atmosphere can affect individual measurements. How did you use wind speed readings to correct for the effect of wind on the readings? Or do you mean you can only discard wind effects if the wind speed is low? In that case, do your data allow to place an upper limit in the wind speed below which its effects can be ignored?

> Yes, we wanted to express that we can exclude wind effects at low wind speeds. We will try to make this clearer in the text. We have not identified a certain wind threshold after which the ML system delivers biased values, however Nolz et al. 2013 reported a three times lower accuracy of lysimeters for a wind speed > 5 m s$^{-1}$. We will rephrase to: "ML data influenced by high wind speed fluctuations (> 5 m s$^{-1}$) could be excluded during such periods to avoid a misinterpretation as water vapor adsorption event. However, during the potential water vapor adsorption period remained below 1 m s$^{-1}$."

Table 1: In the two right-most columns, are the signs of the table entries reversed if the visibility exceeds 1000 m and the temperature is above freezing, respectively? The minus sign could be interpreted as leading to water loss from the ML, but it only signals an absence of the corresponding mode of water deposition. Perhaps explain this in the table heading.

> We will add this information according to your suggestions.

L393: Do you know what effect the closed lysimeter bottom has on the temperature profile inside the ML, compared to in situ values? Also, the ML was 4 degrees warmer at the end of the afternoon (Fig. 5), which you discuss in detail later on. Were you able to determine the cause of the temperature difference? Correction: I see that you discuss this later on.

> Thanks for having taken note of our discussion on this important aspect.

L426-436: The MLs had a lower soil water content than the surrounding soil. You state that this did not affect the NRW. However, it does affect the level of water stress experienced by the plants. In combination this leads to the conclusion that MLs can be used to measure NRW as long as the difference in water stress inside and outside the ML does not lead to changes in soil temperature, canopy architecture, plant height, etc., but cannot be used to study the effect of NRW on the water stress of the vegetation. For that you need deeper lysimeters. Is this correct?

> Yes, this is correct, deeper and wider (greater diameter) lysimeters are necessary to minimize such artefacts, but also normal size lysimeters (taking on the order of one ton of soil) face the oasis problem that conditions inside a lysimeter are never perfectly equal to conditions in undisturbed soils. With respect to our ML system that aims at resolving small NRW inputs, we will add: "The ML system can be used to measure NRW inputs as long as the difference in soil moisture during prolonged drought periods does not influence plant height or canopy architecture.". However, we think that a ML system might still be useful to study the effect of NRW on water stress. The ML system can be used to detect NRW inputs. Plant water stress measurements can be done next to the MLs. During destructive plant water stress measurements, e.g. water potential measurements with a pressure bomb, it is anyways considered to measure in the field in order to avoid manipulation of ML vegetation.

Section 3.6: You present many numbers in the text, which is rather tedious. This information can better be organized in tables.

> We will collect these values in a new table to reduce numbers reported in the text.

Section 4.1: I believe you mean resolution instead of accuracy.

In our understanding accuracy is "how close a reported measurement is to the true value being measured." (https://www.opto22.com/support/resources-tools/demos/accuracy-vs-resolution). The reported measurements are in our case the readings from the microcontroller and the true values stem from the calibration mass as they are used to calibrate commercial scales in grocery stores and elsewhere. We define the term "accuracy" in the introduction: "In this study, weighing accuracy denotes the difference between the measured mass (determined with a ML) and the control (calibrated mass).". "Resolution is the smallest change that can be measured." (https://www.opto22.com/support/resources-tools/demos/accuracy-vs-resolution). So, resolution must be smaller than the accuracy, we report a resolution of 0.0002 mm. In the text we will carefully check the correct usage of accuracy vs. precision. Where a calibration with calibrated mass was possible we will retain the term accuracy. In cases where an absolute standard could not be used, we will adopt the term "precision" instead.

[Figure]

Source: https://www.opto22.com/support/resources-tools/demos/accuracy-vs-resolution

L505: stable decimal place: the meaning of this depends on the units you choose, which you specify elsewhere. I think it is better to rephrase and state the resolution you achieved, compared to that of earlier instruments.

We will delete the sentence to avoid confusion. The resolution is described later on in the same paragraph.

L516: According to the dimension (L105), the ML pots have a volume of 67 liters. They cannot possibly weigh not even 20 kg at that size.

The dimension of the ML pots is 25 cm in diameter and 25 cm in depth (see lines 160–161). Converted to meters and calculating the volume yields $(0.25/2)^2 \times \pi \times 0.25 = 0.0122719$ m$^3$, which corresponds to the 12.2 liters that we report. All three ML had a mass under 20 kg on the order of 15 kg. Maybe you calculated the 67 liters with the dimensions of the outer part (45 cm in diameter x 47.2 cm in depth). The outer part is to protect the inner part and is not the ML pot.

L521: I have the impression this confusion in terminology also appears in this paper.

> We defined these terms in the introduction and tried to stick to this definition throughout the manuscript. "In this study, weighing accuracy denotes the difference between the measured mass (determined with a ML) and the control (calibrated mass). Precision reflects the reliability of the measurements, and it specifies to what extent the experiment can be repeated. On the other hand, resolution is the smallest distinguishable unit for an observable change in mass and thus determines the upper limit of precision." In the revisions we will take great care to rectify potential conflicts between the terms "accuracy" and "precision" as mentioned in earlier responses.

L550: The grass, not the grasslands, grow.

> We will change it to: "The highest NRW inputs occurred during the months of main grass growth (April–September), indicating a potential hydro ecological relevance."

L550: The claim that NRW is highly relevant is a bit too fast. To validate that claim you have to show that it can substantially reduce water stress and/or significantly increases actual transpiration.

> We will rephrase to: "…indicating a potential hydro ecological relevance". In fact, local farmers concluded that the surprising October grass harvest (after a drought with very little rain only terminating the drought), so we probably somewhat overemphasised this aspect based on local understanding of farmers, and thus a more neutral wording is indeed a good idea.

L558-559: 'However, the NRW inputs of the potential water vapor adsorption events were with < 1 mm' I do not understand this, please rephrase.

> We will rephrase to: "However, the NRW inputs of the potential water vapor adsorption events were rather low (0.03 – 0.13 mm)."

L588: But you did not measure the plant mass (impossible to do non-destructively) or the leaf area index, so you may, in fact, have had reduced plant growth that you did not see.

> We will change to: "In this study, this had however no influence on plant standing height because measurements of plant height (before the drought period) and measurement of overall vegetation height (after the drought period) were not statistically different."

L625: ...on plants... Just above you limit yourself to grass. I believe you demonstrated that your system works for short vegetation. For high plants (Maize, shrubs) I am less sure. Also, for vegetation with interlocking leaves, or plants that can be flattened by wind (e.g., barley) and then get back up again, your very sensitive mass measurements may be compromised. Later you claim your system works for plants up to 120 cm. What is the rationale for this value?

> We will rephrase: "Thus, we conclude that our novel ML design is suitable for quantifying nocturnal NRW inputs on grasses and forbs in grasslands reliably and accurately at high temporal resolution.". The claim for up to 120 cm comes from another site, where the grass was that high. However, this other site was not part of this study, hence we will rephrase: "This ML size allowed natural plant growth and

such a ML system can therefore be used in different ecosystems with most short to mid-size statured grasses and forbs or similar vegetation up to ca. 40 cm."

L689: You reported a diameter of 45 cm above, yet here you state that its area is comparable to 25 by 25 cm, which is 0.4 times your ML-area.

The 45 cm are the dimensions of the outer part: "The outer part (Fig. 1a) was made by a cylindrical PVC-U tube (VINK Schweiz GmbH, Dietikon, Switzerland; od45 cm x h42 cm x id44.64 cm) with an open top and a closed bottom." (lines 153-155). The dimension of the ML pot is 25 cm in diameter by 25 cm in depth: "The ML pot was made of a cylindrical PVC-U tube (VINK Schweiz GmbH, Dietikon, Switzerland; od25 cm, h25 cm, id24.8 cm)" (lines 160-163). We realize that our abbreviations: "For better readability, abbreviations for dimensions were used before the corresponding value (d for diameter, h for height or depth, t for thickness)." (lines 150-151) can easily be misunderstood and hence in the revision we will consistently use "25 cm diameter x 25 cm depth".

L694: 'simulate' is not really the right word here. 'Represents', 'reproduce', or 'mimick' are all better, depending on what you want to convey.

We will rephrase and use "represents".

Section 5: You leave out the discrepancy in soil temperatures inside and outside the ML, but it worries me. The temperature difference affects the heat balance of the soil. Liquid water is less viscous in warmer soil, so the hydraulic conductivity increases for a given water content, which will have an effect on the vertical distribution of water and water uptake by roots. A change in the soil temperature affects the partitioning of the incoming energy between heating up the soil and generating evaporation. It also changes the microclimate near the soil surface. Even if the temperature difference vanishes at night, its effects on the soil hydrology may linger. I cannot offer a remedy, and I do not believe it invalidates your measurements, but it is an issue that deserves attention and hopefully can be improved if you continue your work.

To better account for this issue, we will add that higher soil temperatures in ML pots could influence hydraulic characteristics of soil water and the heat balance of the soil which in consequence could lead to biased latent and sensible heat fluxes. We will suggest that further studies should primarily focus to get rid of soil temperature differences between ML pots and the surrounding soil.

Appendix A: It will be challenging to measure drainage with the accuracy necessary for reliable NRW quantification. A few droplets in the outlet tube may have a sizeable effect on the estimated NRW. This appendix is too detailed and wordy. Please condense it to get the message across better.

We completely agree on this aspect of reduced accuracy (see our response to the commenter, Dr. Groh). We provided this information because some anonymous reviewer of an earlier manuscript version insisted on this. The sensor could be placed in an outlet tube with a very steep angle or a PTFE outlet tube (very low friction coefficient). This must be further developed, as we detail in our response to Dr. Groh. In the appendix we provide information on water drainage rates of a ML pot of given

size and provide potential solutions. The potential solutions must be tested and adapted if necessary. We will shorten appendix A by moving numbers to a table and by deleting some sentences.

Jones, H. G.: Monitoring plant and soil water status: established and novel methods revisited and their relevance to studies of drought tolerance, J. Exp. Bot., 58(2), 119–130, doi:10.1093/jxb/erl118, 2006.

Nolz, R., Kammerer, G. and Cepuder, P.: Interpretation of lysimeter weighing data affected by wind, J. Plant Nutr. Soil Sci., 176(2), 200–208, doi:10.1002/jpln.201200342, 2013.

Stiehl-Braun, P. A., Hartmann, A. A., Kandeler, E., Buchmann, N. and Niklaus, P. A.: Interactive effects of drought and N fertilization on the spatial distribution of methane assimilation in grassland soils, Glob. Chang. Biol., 17(8), 2629–2639, doi:10.1111/j.1365-2486.2011.02410.x, 2011.

---

## Author Comment (AC3)

Community comment 1:

Our response to commenter's critique is given in blue indented text after each point addressed by the commenter.

> We thank the commenter Dr. Groh for the comments. Our response to commenter's critique is given in blue indented text after each point addressed by the commenter.

The authors present a new ML system and show how this ML system can be used to determine NRW for a grassland site. The manuscript describes partially very detailed the technical set-up of the ML system. The authors describe that the ML system doesn´t include observations of the ML outflow/drainage and argue in line 207 the ML design was used here to quantify NRW inputs during dry spells and drought periods in summer. I agree that under this conditions the used assumption of no outflow from the ML system might be partially correct. It is also nice to see that the authors recommend in the same section that an ML system would need, when using it in a more general way e.g. to describe NRW inputs for longer time, and additional sensor to determine drainage outflow.

> The commenter is right that our system aims at quantifying the NRW inputs during periods without rain. It is however a misunderstanding, if the commenter understood the wording "more universal" to represent "longer term". The proposed ML system is well suitable for long term measurements, and of course a long-term sum obtained from such a system is a sum in the domain of conditional statistics (i.e. in this case: during dry periods when NRW input can be expected to become a relevant contribution to the hydrological budget). The Covid-19 pandemy has brought about many conditional statistical sums, so we think readers are able to deal with conditional statistics. However, to become even more explicit and clear, we will adopt our text to clarify that our estimate should be considered a conservative NRW input estimate if rainfall periods are included in the averaging, because when doing so (and as specified in Eq. 1), one would simply have to assume zero NRW input during periods that do not qualify for selection as detailed in Section 2.2.8. The critical points outside the scope of our paper given in these lines would then also be the transition periods, shortly after rainfall inputs, e.g. during nights when the sky clears after a rainfall event, NRW could then be underestimated, hence we will clarify in the text that our longer-term estimates of NRW inputs are conservative estimates. We will add more examples to explain what we mean by "more universal" to avoid any confusion. And we will add "during rainfree periods" in the Abstract and Introduction to clarify that the goal is not to obtain better estimates during rainfall periods.

> Our goal was to quantify NRW inputs during drought periods (lines 36–39). We now learn from this comment that this was not clearly enough stated, but by explicitly mentioning this in the Abstract and again in the Introduction (at the bottom where our main objectives are presented), this misunderstanding can be solved.

> Going out of the topic of the paper under discussion we have to remind the commenter that we were unable to suggest a ML system that would also resolve the NRW inputs that could occur during rainfall events with an accuracy that is comparable to that what the proposed ML system can provide under dry conditions. There are four reason why this is a challenge of itself and would have to be solved by someone interested in these inputs at times when there are non-dry conditions in a separate study:

> During rainfall events there is the issue of the splashing of droplets off the ground; a clear definition would be required of how to separate splash droplet water gains from

other water gains; should this then be counted as NRW input (most likely not!), and how could it be distinguished from ground fog water inputs, condensation and adsorption?

Fog droplet inputs under rainfall conditions are also a difficult aspect that would require a special definition. Often during rainfall a near-surface layer of foggy air establishes. Via stable isotopes it can clearly be shown that the fogwater comes from the concurrent rainfall, but once the rainfall stops the fog layer may still be present; should one now count fog droplet input during rainfall and after it has stopped both as NRW inputs? Or only the amount gained after rainfall has stopped? Or should both be considered a secondary pathway of normal rainfall input? In any case a more detailed assessment of the terminology would be required – which is not necessary if one focuses on drought conditions (knowing that by definition we do not count the first minutes and hours after a rainfall event a "drought").

During rainfall events the wetting of the outer side of the ML weighing pot becomes important. While one could quantify the water amount adsorbed to the outer side of the weighing pot in the laboratory, we would expect an increasingly higher weight of this water pool under real-world outdoor conditions because of e.g. algae growing on the outer face of the ML pot, and consequently also an increasing share of accumulated hydrophibic soil particles, etc.

No matter how one would chose the definitions for separating the components, the experimental errors to quantify "NRW" inputs during and (shortly) after rainfall events would explode. Currently we could only think of a system that would be 5–10 times less accurate than what we present for drought conditions.

This is not to say that it is impossible to achieve high-quality NRW input data during rainfall events, but it is to explain why we did not include this aspect in our project. By focusing on drought periods we circumvented all these technical challenges, well knowing that during rainfall events NRW inputs may not be perfectly zero (as our approach assumes, see Eq. 1), hence our suggestion to explicitly mention that our estimates are conservative estimates if rainfall periods are included in the total.

Thus I don't understand why the authors showing in section 3.6 NRW inputs over one year here, when the know that the ML system cannot provide such data? I recommend to delete the section 3.6 as the shown total amounts are strongly biased by the inability of the ML system to correctly quantify NRW input during time were also drainage occurs.

For the analysis of the NRW input for one year we excluded periods when rainfall occurred as mentioned above. We will add the information that this is a "conservative estimate" and the reason of course is still that we aimed at quantifying the NRW input during drought periods when there is a potential for this component to become relevant in the (dry period) hydrological budget. This is of course a conditional total, as if you sum up rainfall of days where the intensity is exceeding a certain threshold. That's sound statistics. We described this in the Methods section (2.2.8): "If rainfall occurred during an analysed 24-hour period, that period was excluded, except the rain event occurred directly after the NRW input event." (lines 308-309). Thus, the ML system was used the determine NRW input during periods with no rainfall. NRW inputs could be underestimated shortly after rainfall events. "Nevertheless, under conditions when drainage water flow persists for a longer time, the ML system

provides conservative estimates of NRW inputs. A possible modification of the ML system to also accurately quantify such drainage flow is suggested in the appendix." (lines 447-449). Please note that our suggestion added to the appendix is to show how this could be done, but in our view we would not get accurate estimates of NRW inputs during rainfall periods as our response above details. Since this aspect is out of scope of our manuscript, we offer this extension as a thought input for anyone who is interested in developing an accurate ML device to quantify NRW inputs during rainfall events. We will make it clearer in the manuscript text that rainfall periods were excluded, and that our ML system might give conservative estimates if drainage water flow persists for longer time. We were of the opinion that the term "non-rainfall water inputs" was clear enough to the reader that this is not including rainfall periods, but we accept this commenter's view that a reader expecting non-rainfall water inputs to be important during rainfall events could be misunderstanding our message, hence we will modify Abstract and Introduction to be more explicit and clear on this aspect that indeed is out of scope of our manuscript.

please show the ML system installed in the field somewhere in the Material and Method section

We will add photographs of the ML system and photographs taken during installation of the ML system.

5 please show day and nighttime here and show in a) more the just the very close vicinity of the ML system in the pictures.

We interpret that with "5" Fig. 5 was meant and not line 5 or section 5. We already show more than the close vicinity of the ML system in Fig. 5b. "To compare ML pot temperatures to temperatures of the surrounding, separate images were taken in a distance of ca. 100 cm (images not shown here) with a size of ca. 75x75 cm, to exclude potential influence of the ML on its approximate surrounding.". We added these separate images for the commenter. Thermal images were taken during 18:27 and 05:15, no daytime images were taken. This ML system was designed to measure NRW inputs during nights.

[Figure]

18:27 UTC

19:46 UTC

20:39 UTC

[Figure]

The dry out periods in Fig.7 a) showing large difference from July until September between different pots and the control. The authors only show this for one period in July but it is also visible in august and September! The argument that nighttime difference are small is not correct as B) shows that only for a very small time window around 6 in the morning the difference is close to zero. However dew starts, as shown in a previous figure (Fig. 5) much earlier at around 7 pm where differences are still large (~2°C shown in Fig. 7 c). Thus the conclusion of the authors that that soil temperatures inside ML pots during the most relevant hours of day when dew forms (during the night before sunrise) from line 429-430 is partially not correct.

This might be a partial misunderstanding. On lines 429-430 we wrote: "From this we conclude that soil temperatures inside ML pots during the most relevant hours of day when dew forms (during the night before sunrise) **were not strongly influenced by a lower water content and its resulting lower heat capacity**." We only conclude that the cause of soil temperature differences were not resulting from a lower WFPS (given that WFPS varied quite strongly among pots). This is what you would expect: had the soil dried out too much, then its temperature measured at 15 cm depth (line 286) would strongly increase, which was not the case (relative comparison), but this does not express that the ML pot soil temperatures are not a bit higher than the reference in the solid soil control plots (as is clearly shown in Fig. 7c). Such temperature differences can only be minimized by maximizing the size of the lysimeter beyond what would be termed a micro-lysimeter (and which is the trade-off for higher accuracy NRW quantification that we aimed at). Hence our information will help future investigators to further improve the system in this and other aspects.

On lines 433–436 we quantify this effect of temperature differences: "Over the period from May till October 2019 (Fig. 7c), the hourly mean soil temperature deviations of ML pot 1 from the control ranged between –0.14 °C around sunrise and 2.57 °C in the later afternoon. Thus, during most of the night when NRW input occurs, the temperature differences between the soil of ML pots and the control are typically less than 1 °C.". We will change the last sentence to: "…90% of nocturnal temperature values were below 2.90 °C and 50% of the values were below 0.69 °C."

Section NRW inputs over one year showing strongly biased NRW data as the ML system fails to correctly quantify NRW under conditions were also drainage occurs and I was very surprise that this topic was even not picked up in the discussion section 4.4.

The reason for this impression by the commenter is that we thought we were clear enough to clarify that we are aiming at NRW inputs during drought periods. We will thus add this aspect to the discussion section 4.4 and clarify that we excluded rainfall periods (as specified in Eq. 1) and that the ML system might give conservative NRW estimates when drainage water outflow occurs (see also first paragraph of the response).

The appendix A: drainage water flow of ML pots is too speculative from my perspective. Drainage occurs not only during rainfall and shortly after rainfall as mentioned in line 792. The outflow from soil depends on their soil characteristics and thus might differ when using ML system at other sites and different soils. The outflow from soils are typically low additional also bias ET during the day!

In the appendix A we used measured data from the ML system and the close-by rain bucket during and after a heavy rainfall event. If drainage persists for a longer time period, then the ML system would give conservative estimates of NRW inputs (as we then set NRW input to zero, see Eq. 1). We will present this argument clearer in the revised manuscript. Furthermore, we will add that the ML system was tested at the Früebüel study site and that soil characteristics at other sites might differ, and thus different patterns of drainage water outflow might occur.

---

## Author Response (AR1)

**Anonymous Referee 1:**

As an important component of the ecohydrological cycle, it is always difficult to analyze and quantitatively characterize Non-rainfall water (NRW). In this paper, the author designed, manufactured and tested a novel micro-lysimeter (ML) system, which has high precision and weighing range, and overcomes some defects of the existing lysimeters. Different types of NRW inputs, such as dew, hoar frost, fog, rime and the combinations among these, are well distinguished by auxiliary sensors. At the same time, it is also similar to the surrounding environment in terms of canopy and soil temperature, plant growth and soil humidity. The author applied the ML system at a field site in Switzerland. Through the monitoring of a hydrological year, different NRW events were effectively distinguished and the NRW inputs was quantified. In general, the paper has good innovation and practical value, but some parts need to be improved, and major revision are suggested. The specific opinions are as follows.

- 1. It is recommended to attach the location map of the study site in Switzerland and the real photos of the ML system;
- 2. How deep is the groundwater level? Should the impacts of groundwater be considered?
- 3. Line 215-220, it may be difficult to transfer the soil body from one ML to the second. How to ensure the stability of soil during transfer?
- 4. Do you think freezing-thawing would have impacts? Should the accuracy of the ML change at different temperatures?
- 5. Can the ML distinguish the influences from dust or other drifting materials falling on it?
- 6. According to the author's statement, the ML can only distinguish windless conditions, so it cannot work on windy days? NRW input over a year in Figure 8 might be underestimated?
- 7. The ML can quickly discharge the excess water after precipitation, but this would also cut off the evapotranspiration channel, which might misrepresent the conditions in the nearby soils. Will it affect NRW?
- 8. In figure 7 (b), it is described as WFPS in the text, but not in this figure.
- 9. Different types of NRW inputs in Figure 8 can be represented in different colors. The same is for Figure 9.

**Response to Referee's comments:**

We thank the anonymous reviewer for his/her constructive feedback and inspiring questions. We did our best to improve our manuscript accordingly. Our response to reviewer's suggestions and critique is given in blue indented text point by point. Line numbers are referring to the clean revised manuscript version.

- 1. We added a location map and an aerial photograph of the site to Appendix A Fig. A1. Furthermore, we added photographs that show the installation process, the uninstalled and the installed ML system to Appendix B Fig. B1.
- 2. Please also have a look at our response in the final author comments. The disconnection of the ML pot to the underlying soil and in consequence to the groundwater body is a general lysimeter problem. We added to (lines 79–82) that there is also a disconnection to the groundwater body. "While limited rainfall retention capacity of ML is not a problem for NRW quantification, the potential prevention of upward direct water flow due to capillary rise from deeper soil layers or

the groundwater body cannot be neglected (Evett et al., 1995), because it replenishes plant available water in the rooting zone."

- 3. Indeed! This is quite a challenge. From our experience the monolith breaks most likely at the bottom, whereas breaking at the edges is not a big issue. At the time of installation, we were three people at the site and could manage to retrieve three undisturbed soil monoliths after one failure. We found that this work is best done when the soil is not completely dry or extremely wet to avoid any breaking of the monolith. We added: "The reversed soil monolith was carefully taken out from the ML pot and three people collaborated to transfer it to a second ML pot to be upright again. The ML pot was then ready for installation on the weighing platform." (lines 729–731). Furthermore, we added photographs of the installation process to Appendix B, Fig. B1 a-d.
- 4. Freezing-thawing could damage PVC parts of the ML, however this did not occur during the study period. We added to the text: "During or after freezing temperature conditions the ML system should be controlled, because expanding water in the reservoir or the ML pot could break PVC parts of the ML system. However, this did not occur during this study period." (lines 488–491). We added that the used load cells were temperature compensated. "Between the load plate and the base plate, a PW15AHY temperature-compensated load cell with 20 kg capacity (HBM, Darmstadt, Germany) was mounted." (lines 154–156).
- 5. Thank you for pointing this out. We added: "In deserts or arid regions (with low vegetation cover) additional sensors (e.g. infrared video cameras) would be needed to detect depositing materials like dust and sand that accumulate on the ML over time." (lines 556–558).
- 6. We added: "However, NRW inputs occur during conditions with low wind speed, the probability for dew formation decreases below 5% when wind speeds are smaller than 0.4 m s-1 or bigger than 1.9 m s-1 (Zhang et al., 2014). Thus, wind is not a big bias source for NRW quantification." (lines 477–479).
- 7. The water source of NRW inputs is atmospheric water vapor. The atmospheric water vapor can also stem from evaporated soil water, termed distillation. Distillation can contribute up to 42 % of the condensed water on foliage during dew nights (Li et al., 2020). Thus, a ML pot that is cut off from the evapotranspiration channel might contribute less to the distillation process. However, in the boundary layer atmospheric water vapor is distributed via turbulence and thus the ML is subject to the same atmospheric water vapor as the surrounding field. This cut-off might still lead to the same amount of condensation (e.g. dew formation on plants). However, prolonged altering of soil moisture could have an influence on plant growth and in consequence to NRW inputs (please have also a look at our response to a similar comment by reviewer 2).
- 8. Thanks for pointing this out. We used once "WFPS" instead of "Soil temperature" which created some confusion. We corrected it.
- 9. We changed the figures, each colour represents now a different type of NRW input.

Li, Y., Aemisegger, F., Riedl, A., Buchmann, N. and Eugster, W.: The role of dew and radiation fog inputs in the local water cycling of a temperate grassland in Central Europe, Hydrol. Earth Syst. Sci. Discuss., 1–27, doi:10.5194/hess-2020-493, 2020.

Zhang, Q., Wang, S., Yang, F. L., Yue, P., Yao, T. and Wang, W. Y.: Characteristics of Dew Formation and Distribution, and Its Contribution to the Surface Water Budget in a Semi-arid Region in China, Boundary-Layer Meteorol., 154(2), 317–331, doi:10.1007/s10546-014-9971-x, 2014.

**Anonymous Referee 2:**

We thank the anonymous reviewer for the constructive and detailed feedback. We did our best to improve our manuscript accordingly. Our changes to reviewer's critique are given in blue indented text after each point addressed by the reviewer. Line numbers are referring to the clean revised manuscript version.

The paper introduces a microlysimeter for the specific purpose of measuring the deposition of water on the surfaces of the soil and the vegetation by means other than rain (dew, fog, etc.). The deposited amounts of water are quite small, yet the paper argues that they could be important for the vegetation during dry periods. The design of the set-up, with its high observation frequency and high-resolution mass measurements is explained in some detail. The instrument has been in operation in the field. Its performance is reported and evaluated.

**General comments**

Overall, the paper gets across the relevance of MLs and the improvements the authors have made to earlier designs. The material fits the HESS mission and its readership.

That being said, the paper is wordy and tedious at times. The authors go in so much detail that is hard to follow the line of thought. In contrast, Figure 1 leaves out many technical details, and photographs of the ML and its components are not given.

We have now modified Fig. 1 to show a photograph of the inside part of the ML as panel (a) and a sketch which shows the dimensions of the components at scale in panel (b). We also added an item q in panel (b) to more clearly show that an additional sensor is required if also drainage water rate should be quantified.

The Introduction is comprehensive but incoherent, and would benefit from a careful revision. It can be shortened a little perhaps.

Thanks for this suggestion. We shortened and completely rearranged the Introduction to increase coherence.

In Materials and Methods, there is no information at all given about the soil of the experimental site. The technical details of the ML setup and its installation and operating procedures need to be described better. The rest of the Methods are very detailed, sometimes about well-established techniques. You can shorten the text there.

We added information about the soil at the site: "The soil at the site is a silt loam mixture (56% silt, 37% sand, 7% clay), with a bulk density of  $1.12 \pm 0.03$  g cm-3 and an organic C content of  $4.4 \pm 0.2\%$  (Stiehl-Braun et al., 2011)." (lines 115–117). We added photographs of the ML system and of the installation process to Appendix B, to better explain the installation and operating procedures. To shorten the Methods, we moved the paragraphs: "Soil monolith preparation" to Appendix B, "Data collection, storage and delivery" to Appendix C, "Load cell data low-pass filtering" to Appendix D. We relegate at the beginning of the Methods to the Appendix, in order that interested readers can find these information.

The authors make a point about separating the various NRW modes, but given the small total flux, I do not understand why this is so important. On the other hand, the differences in soil temperature between the ML and the surrounding soil is downplayed, even though its various effects may linger into the night, when NRW occurs.

In accordance with the Editor (please have a look at the Editor report) we retained the part about separating various NRW inputs and added more explanation why this is important for the readership with another background: "Differentiation among different types of NRW inputs is important for various research disciplines, e.g. the prediction of fog events poses a major challenge for numerical weather prediction for meteorologists (Westerhuis et al., 2020). Thus, it is important to measure the frequency and water inputs of fog events during the whole year." (lines 535–538). For the difference in soil temperature, please have a look at the response on the comment to Section 5.

The material on which the paper is based is solid, but the presentation is not so good. The paper would benefit from a thorough rewrite that increases its coherence and clarity, and reduces its size a little.

We tried to increase to coherence of the paper by rearranging and deleting some parts.

The paper mentions a supplement but I could not find that, other than the data set.

We apologize for this error, the supplement has been included as an Appendix, but we overlooked the need for a change in wording. We changed "Supplement" to "Appendix".

**Detailed comments**

L34: The sentence introducing the new ML looks out of place here. You are still developing the argument for its necessity, only moving from general terms to a specific ecosystem.

We agree and rearranged the Introduction.

L39: If rainfall ('RW') is absent, NRW necessarily IS the only atmospheric source of water, because RW and NRW are mutually exclusive and complementary.

We deleted this sentence while rearranging the Introduction.

L43: 'another temperate site' You did not mention the first one.

We deleted this sentence while rearranging the Introduction.

**L51: What do you mean by plant water status?**

We tried to clarify this term by adding: "Plant water status is a widely used measure in plant physiology for assessing plant water stress. It incorporates the amount of water in plants and its energy status (Jones, 2006). NRW inputs can increase the amount of water in plants (Limm et al., 2009; Munné-Bosch and Alegre, 1999) and increase thereby the plant water status, which lowers plant water stress." (lines 39– 43).

**L56 and 320: You mention dew deposition on soil, but earlier you stated that does rarely occur.**

It does rarely occur, but still there is the possibility that it occurs. Thus, we would prefer to keep this sentence.

You need to have a careful look at the Introduction. Although I agree with the arguments it presents, they are presented in a confusing order. Above I mentioned the Introduction of a lysimeter in the middle of a discussion about NRW. Elsewhere too you jump between general statements and location-specific arguments without a logical connection between the two. This compromises the coherence of the text and disrupts the flow of thought. All the elements the Introduction needs are there, but please present them in a more coherent order and use more paragraphs as compartments for different focal points of the Introduction.

We rearranged the Introduction as recommended by the reviewer and deleted and shortened where necessary and meaningful.

Section 2: Some subsections are not directly important to understand the ML setup. Perhaps the very detailed material can be placed in the supplement.

We moved three rather detailed subsections to the Appendix: 2.2.3 Soil monolith preparation (now Appendix B); 2.2.4 Data collection, storage and delivery (now Appendix C); 2.2.5 Load cell data low-pass filtering (now Appendix D).

**Section 2.1: Please give some information about the soil.**

We added: "The soil at the site is a silt loam mixture (56% silt, 37% sand, 7% clay), with a bulk density of  $1.12 \pm 0.03$  g cm-3 and an organic C content of  $4.4 \pm 0.2\%$  (Stiehl-Braun et al., 2011)." (lines 115–117).

**L124: What is FluxNet? Do we need to know?**

This is part of the site information to tell readers in which network this site is embedded. Below is a statistics extracted from the Scopus database on the increasing importance of FluxNet in the field of eddy covariance flux measurements of  $H_2O$  and  $CO_2$  fluxes. It is thus not surprising that the name is new to some readers, but this might change rapidly if the spread of FluxNet data usage continues at its current pace. Thus, we prefer to keep this information and expect that readers not interested in it will simply ignore.

L154: Instead of the width and the thickness it is probably better to give the outer and inner diameter of the ML tube.

We changed it according to your suggestions. We use now outer diameter and inner diameter. Furthermore, we realized that our used abbreviations created some confusion, thus we have written out dimensions, e.g. we use now "diameter" instead of "d".

L173-174: How do you level the load cell and the ML during installation in the field? I cannot see the adjustment screws in the figure or how you can reach them during the installation process.

The adjustment screws (=adjustable support feet) are shown in Fig. 1b as item "h". We added a further component Fig. 1b:i counter nut for adjustable support feet to explain the "H" like structure better. We added labels for both visible support feet. Furthermore, we added a photograph of the weighing platform (Fig. 1a) and the adjustable support feet, which are basically machine screws (bolts) with a counter nut (otherwise the weighing platform would move). We could reach them with a prolonged hexagon socket wrench. We added to Appendix B: "The weighing platform was levelled out by adjusting the three adjustable standing feet with a prolonged hexagon socket wrench. The final position was fixed with the counter nut by using an open end wrench." (lines 731–733).

Fig. 1: What are the structures on either side of the load cell that look like an H on its side? The text mentions machine screws (bolts?) but I cannot find these in the figure. I also think it would be helpful to add a photograph of the instrument to the figure, as well as a scale or a set of dimensions within the drawing.

Please have a look at the response to L173-174.

Section 2.2.3: I estimate a filled ML weighs about 100 kg. How did you handle it when you excavated the monolith and when you installed the monolith in its field setup?

The ML mass was lower than 20 kg for all three ML (ML pot size of 25 cm diameter x 25 cm depth). The maximum capacity of the load cell was also 20 kg. We think that our used abbreviations in the Methods section (e.g. "d" instead of diameter) created some confusion, we hope that the dimensions are now better understandable. At the day of installation, we were three people at the site, and we were able to transfer the monoliths to the ML pots and afterwards the whole weighing platform to the outer part. For heavier weights one would need a crane, we agree, but with this weight it is not an issue. We added: "The reversed soil monolith was carefully taken out from the ML pot and three people collaborated to transfer it to a second ML pot to be upright again. The ML pot was then ready for installation on the weighing platform." (lines 729–731).

The ML pots were closed on one side. In order to transfer the monolith from the sampling pot to its ML pot you need to place both pots with the open sides against each other, so you end up with a cylinder that is closed on both sides. How did you transfer the monolith from one half of that cylinder to the other?

We added some photographs and a description to Appendix B Fig. B1, we think that helps to understand this process better. Basically, we took it out from one ML pot and transferred it by hand, in upright position, to another ML pot.

**L240: What was the size of the averaging window?**

We added that the size of the averaging window was 100 values (this was the maximum that this microcontroller could handle). Thus, with a sampling frequency of 3.3 Hz, the averaging window had a length of 30.3 seconds. We added: "The raw load cell data were then stored in an averaging window (ring memory) with a size of 100 values, where the oldest values were replaced by the newest ones." (lines 753–755).

**Section 2.2.6: it is helpful to mention here somewhere what the resolution of the load cell is converted to mm water layer.**

This information is not available. The load cell itself outputs only a voltage, that's why also the company does not provide such a measure. The resolution depends on the parts of the system, e.g. if one uses a 8-Bit or a 16-Bit analog digital converter, the calibration range and many other factors. We provide information about the resolution of our system in Section 3.1, thus in the results section, not in the methods section, because this is a setup-dependent value. Here we specify the SE of the measurements of the three ML as  $\pm 0.31$ ,  $\pm 0.14$  and  $\pm 0.11$  g, respectively. We added the lacking units in Section 3.1 and Fig. 2.

**L279: Why the**

L327: Accuracy of what? In the section that follows you use the term accuracy a lot, but I believe you sometimes mean precision (e.g., https://www.mccdaq.com/TechTips/TechTip-1.aspx).

We changed it to: "Accuracy of the ML system" (line 463). In the text we will carefully check the correct usage of accuracy vs. precision. We moved the definition of the terms accuracy, precision and resolution from the Introduction to the Methods section. We tried to better explain the term "precision" when using it: "Precision (repeatability of the measurements)..." (line 492).

L328: Please give more significant digits for the correlation coefficient.

We provide now four positions after decimal point: "Three replications showed an almost perfect linear correlation ( $R^2=0.9999$ ) between target mass and load cell mass." (lines 292–293).

L365: with your measurement frequency, individual eddies in the near-surface atmosphere can affect individual measurements. How did you use wind speed readings to correct for the effect of wind on the readings? Or do you mean you can only discard wind effects if the wind speed is low? In that case, do your data allow to place an upper limit in the wind speed below which its effects can be ignored?

Yes, we wanted to express that we can exclude wind effects at low wind speeds. We tried try to make this clearer in the text by simply stating in the Results section: "Wind speed remained low ( $< 1 \text{ m s}^{-1}$ ) during the whole potential water vapor adsorption event." (lines 329–330) and by stating in the discussion: "Ancillary wind measurements could be used to exclude periods with high wind speeds, because high wind could act as a force on ML and increase thereby mass. However, NRW inputs occur during conditions with low wind speed, the probability for dew formation decreases below 5% when wind speeds are smaller than 0.4 m s-1 or bigger than 1.9 m s-1 (Zhang et al., 2014). Thus, wind is not a big bias source for NRW quantification." (lines 475–479).

Table 1: In the two right-most columns, are the signs of the table entries reversed if the visibility exceeds 1000 m and the temperature is above freezing, respectively? The minus sign could be interpreted as leading to water loss from the ML, but it only signals an absence of the corresponding mode of water deposition. Perhaps explain this in the table heading.

We added this information according to your suggestions: "The sign '+' indicates the presence, whereas the sign '-' indicates the absence of a certain factor." (lines 350–351).

L393: Do you know what effect the closed lysimeter bottom has on the temperature profile inside the ML, compared to in situ values? Also, the ML was 4 degrees warmer at the end of the afternoon (Fig. 5), which you discuss in detail later on. Were you able to determine the cause of the temperature difference? Correction: I see that you discuss this later on.

**Thanks for having taken note of our discussion on this important aspect.**

L426-436: The MLs had a lower soil water content than the surrounding soil. You state that this did not affect the NRW. However, it does affect the level of water stress experienced by the plants. In combination this leads to the conclusion that MLs can be used to measure NRW as long as the difference in water stress inside and outside the ML does not lead to changes in soil temperature, canopy architecture, plant height, etc., but cannot be used to study the effect of NRW on the water stress of the vegetation. For that you need deeper lysimeters. Is this correct?

Yes, this is correct, deeper and wider (greater diameter) lysimeters are necessary to minimize such artefacts, but also normal size lysimeters (taking on the order of one ton of soil) face the oasis problem that conditions inside a lysimeter are never perfectly equal to conditions in undisturbed soils. With respect to our ML system that aims at resolving small NRW inputs, we added: "Thus, the ML system can be used to reliably measure NRW inputs as long as the difference in soil moisture during prolonged drought periods does not influence plant height or canopy architecture." (lines 573–575). However, we think that a ML system can be used to study the effect of NRW on water stress. The ML system can be used to detect NRW inputs. Plant water stress measurements can be done next to the MLs. During destructive plant water stress measurements, e.g. water potential measurements with a pressure bomb, it is anyways considered to measure in the field in order to avoid manipulation of ML vegetation.

**Section 3.6: You present many numbers in the text, which is rather tedious. This information can better be organized in tables.**

We collected these values in a new table "Table 2" to reduce numbers reported in the text (lines 425 onwards).

**Section 4.1: I believe you mean resolution instead of accuracy.**

In our understanding accuracy is "how close a reported measurement is to the true value being measured." (https://www.opto22.com/support/resourcestools/demos/accuracy-vs-resolution). The reported measurements are in our case the readings from the microcontroller and the true values stem from the calibration mass as they are used to calibrate commercial scales in grocery stores and elsewhere. We define the term "accuracy" in the Methods section: "In this study, weighing accuracy denotes the difference between the measured mass (determined with a ML) and the control (calibrated mass)." (lines 231–232) "Resolution is the smallest change that can be measured." (https://www.opto22.com/support/resources-tools/demos/accuracy-vs-resolution). So, resolution must be smaller than the accuracy, we report a resolution of 0.0002 mm. L505: stable decimal place: the meaning of this depends on the units you choose, which you specify elsewhere. I think it is better to rephrase and state the resolution you achieved, compared to that of earlier instruments.

We deleted this sentence to avoid confusion. The resolution is described later on in the same paragraph.

L516: According to the dimension (L105), the ML pots have a volume of 67 liters. They cannot possibly weigh not even 20 kg at that size.

The dimension of the ML pots is 25 cm in diameter and 25 cm in depth (see lines 160–161). Converted to meters and calculating the volume yields  $(0.25/2)^2 \times \pi \times 0.25 = 0.0122719 \text{ m}^3$ , which corresponds to the 12.2 liters that we report. All three ML had a mass under 20 kg on the order of 15 kg. Maybe you calculated the 67 liters with the dimensions of the outer part (45 cm in diameter x 47.2 cm in depth). The outer part is to protect the inner part and is not the ML pot. We adapted the description of the dimensions in the methods Section 2.2.1 and hope thereby to have eliminated confusion about dimensions.

**L521: I have the impression this confusion in terminology also appears in this paper.**

We defined these terms in the Methods section and tried to stick to this definition throughout the manuscript. "In this study, weighing accuracy denotes the difference between the measured mass (determined with a ML) and the control (calibrated mass). Precision reflects the reliability of the measurements, and it specifies to what extent the experiment can be repeated. On the other hand, resolution is the smallest distinguishable unit for an observable change in mass and thus determines the upper limit of precision." (lines 231–234).

**L550: The grass, not the grasslands, grow.**

We changed it according to your suggestion: "The highest NRW inputs occurred during the months of main grass growth (April–September), indicating a potential hydro ecological relevance." (lines 533–534).

L550: The claim that NRW is highly relevant is a bit too fast. To validate that claim you have to show that it can substantially reduce water stress and/or significantly increases actual transpiration.

We rephrased to: "...indicating a potential hydro ecological relevance" (line 534). In fact, local farmers concluded that the surprising October grass harvest (after a drought with very little rain only terminating the drought), so we probably somewhat overemphasised this aspect based on local understanding of farmers, and thus a more neutral wording is indeed a good idea.

L558-559: 'However, the NRW inputs of the potential water vapor adsorption events were with < 1 mm' I do not understand this, please rephrase.

We rephrased to: "However, the NRW inputs of the potential water vapor adsorption events were rather low (0.03 - 0.13 mm)." (lines 544–545).

L588: But you did not measure the plant mass (impossible to do non-destructively) or the leaf area index, so you may, in fact, have had reduced plant growth that you did not see.

We changed it to: "In this study, this had however no influence on plant standing height because measurements of plant height (before the drought period) and measurement of overall vegetation height (after the drought period) were not statistically different." (lines 569–571).

L625: ...on plants... Just above you limit yourself to grass. I believe you demonstrated that your system works for short vegetation. For high plants (Maize, shrubs) I am less sure. Also, for vegetation with interlocking leaves, or plants that can be flattened by wind (e.g., barley) and then get back up again, your very sensitive mass measurements may be compromised. Later you claim your system works for plants up to 120 cm. What is the rationale for this value?

We rephrased: "Thus, we conclude that our novel ML design is suitable for quantifying nocturnal NRW inputs on grasses and forbs reliably and accurately at high temporal resolution." (lines 608–610). The claim for up to 120 cm comes from another site, where the grass was that high. However, this other site was not part of this study, hence we will rephrase: "This ML size allowed natural plant growth and such a ML system can therefore be used in different ecosystems with most short to mid-size statured grasses and forbs or similar vegetation up to ca. 40 cm." (lines 681–683).

L689: You reported a diameter of 45 cm above, yet here you state that its area is comparable to 25 by 25 cm, which is 0.4 times your ML-area.

The dimension of the ML pot is 25 cm in diameter by 25 cm in depth. Our wording "25 x 25 cm" can easily be misunderstood, thus we changed it to: "25 cm diameter  $\times$  25 cm depth" (line 680).

L694: 'simulate' is not really the right word here. 'Represents', 'reproduce', or 'mimick' are all better, depending on what you want to convey.

We rephrased and used "represents" (line 684).

Section 5: You leave out the discrepancy in soil temperatures inside and outside the ML, but it worries me. The temperature difference affects the heat balance of the soil. Liquid water is less viscous in warmer soil, so the hydraulic conductivity increases for a given water content, which will have an effect on the vertical distribution of water and water uptake by roots. A change in the soil temperature affects the partitioning of the incoming energy between heating up the soil and generating evaporation. It also changes the microclimate near the soil surface. Even if the temperature difference vanishes at night, its effects on the soil hydrology may linger. I cannot offer a remedy, and I do not believe it invalidates your measurements, but it is an issue that deserves attention and hopefully can be improved if you continue your work.

To better account for this issue, we added: "Soil temperatures were higher in ML pots, especially during the day. This could influence the hydraulic characteristics of soil water, the heat balance of the soil and in consequence lead to biased latent and

sensible heat fluxes. Thus, further ML studies should primarily focus to get rid of soil temperature differences between ML pots and the surrounding soil." (lines 692–696).

Appendix A: It will be challenging to measure drainage with the accuracy necessary for reliable NRW quantification. A few droplets in the outlet tube may have a sizeable effect on the estimated NRW. This appendix is too detailed and wordy. Please condense it to get the message across better.

We completely agree on this aspect of reduced accuracy (see our response to the commenter, Dr. Groh). We provided this information because some anonymous reviewer of an earlier manuscript version insisted on this. The sensor could be placed in an outlet tube with a very steep angle or a PTFE outlet tube (very low friction coefficient). This must be further developed, as we detail in our response to Dr. Groh. In the Appendix we provide information on water drainage rates of a ML pot of given size and provide potential solutions. The potential solutions must be tested and adapted if necessary. We shortened (now Appendix E) by moving numbers to a table and by deleting some sentences.

Jones, H. G.: Monitoring plant and soil water status: established and novel methods revisited and their relevance to studies of drought tolerance, J. Exp. Bot., 58(2), 119–130, doi:10.1093/jxb/erl118, 2006.

Limm, E., Simonin, K., Bothman, A. and Dawson, T.: Foliar water uptake: a common water acquisition strategy for plants of the redwood forest, Oecologia, 161(3), 449–459, doi:https://doi.org/10.1007/s00442-009-1400-3, 2009.

Munné-Bosch, S. and Alegre, L.: Role of Dew on the Recovery of Water-Stressed Melissa officinalis L. Plants, J. Plant Physiol., 154(5–6), 759–766, doi:10.1016/S0176-1617(99)80255-7, 1999.

Nolz, R., Kammerer, G. and Cepuder, P.: Interpretation of lysimeter weighing data affected by wind, J. Plant Nutr. Soil Sci., 176(2), 200–208, doi:10.1002/jpln.201200342, 2013. Pedregosa, F., Varoquaux, G., Gramfort, A., Michel, V., Thirion, B. and Grisel, O.: Scikitlearn: Machine learning in Python, J. Mach. Learn. Res., 12(Oct), J. Mach. Learn. Res., 2011.

Stiehl-Braun, P. A., Hartmann, A. A., Kandeler, E., Buchmann, N. and Niklaus, P. A.: Interactive effects of drought and N fertilization on the spatial distribution of methane assimilation in grassland soils, Glob. Chang. Biol., 17(8), 2629–2639, doi:10.1111/j.1365-2486.2011.02410.x, 2011.

Westerhuis, S., Fuhrer, O., Cermak, J. and Eugster, W.: Identifying the key challenges for fog and low stratus forecasting in complex terrain, Q. J. R. Meteorol. Soc., 146(732), 3347–3367, doi:10.1002/qj.3849, 2020.

Zhang, Q., Wang, S., Yang, F. L., Yue, P., Yao, T. and Wang, W. Y.: Characteristics of Dew Formation and Distribution, and Its Contribution to the Surface Water Budget in a Semi-arid Region in China, Boundary-Layer Meteorol., 154(2), 317–331, doi:10.1007/s10546-014-9971-x, 2014.

**Community comment 1:**

We thank the commenter Dr. Groh for the comments. Our response to commenter's critique is given in blue indented text after each point addressed by the commenter. Line numbers are referring to the clean revised manuscript version.

The authors present a new ML system and show how this ML system can be used to determine NRW for a grassland site. The manuscript describes partially very detailed the technical set-up of the ML system. The authors describe that the ML system doesn't include observations of the ML outflow/drainage and argue in line 207 the ML design was used here to quantify NRW inputs during dry spells and drought periods in summer. I agree that under this conditions the used assumption of no outflow from the ML system might be partially correct. It is also nice to see that the authors recommend in the same section that an ML system would need, when using it in a more general way e.g. to describe NRW inputs for longer time, and additional sensor to determine drainage outflow.

Thanks for the critical comment, but we respectfully have to disagree in aspects and hence decided to add a full-fledged Appendix F to simulate drainage water flow from the bottom of the soil monolith. We did **not** make the assumption that there is no drainage after rainfall events, but we emphasized that we are not quantifying NRW inputs during times when there might be drainage flow. Thus, we set NRW inputs = 0 mm during rainfall periods and the time thereafter until dry and drought conditions are observed on which our focus is. Thus, **our assumption is that NRW inputs are negligible during and shortly after rain events** given the orders of magnitude higher precipitation rates than NRW input rates during dry and drought conditions. We are sorry that our wording was not understandable to the commenter and hence have rephrased more explicitly.

In Appendix F with Fig. F1 and Table F1 we set up a model following Zhan et al. (2016) and simulated the duration td until less than one drop of water per square meter and day (!) is draining at the bottom of a monolith. As the starting condition we chose fully saturated soils, which in reality requires more than 5 days of substantial aboveaverage precipitation (!). Fig. F1 then shows the range for different soil types. At our field site we have a silt loam, which yields a best estimate of  $t_d = 7.2$  hours; but when we tried to explore the full range of td with parameter combinations within the ranges given by Rawls et al. (1991) that we present in Table F1, the maximum td can be up to 33.8 hours with a two-sided 90% (80%, 50%) CI of 0.0–24.8 hours (0.0–20.3 hours, 0.0–12.2 hours). Because soils typically do not fully saturate during normal precipitation events, we also show simulated  $t_d$  where we measure the time when the average soil water content of the monolith passes 90%, 80%, and 70% of its saturation value. The corresponding  $t_d$  (see Fig. F1) are: 5.0 hours, 4.2 hours, and 3.5 hours, respectively. Thus, we are convinced that eliminating the 24 hours after rainfall in our analysis is a realistic approach to (a) really look at periods of dry or drought conditions, and (b) to support our approach that during dry and drought conditions we do not require a measurement of drainage flow at the bottom of the monolith. This modeling revealed to be quite a bit more challenging than expected, and thus we obtained the help of a Ph.D. student in mathematics, which helped to solve issues and supported WE with the model set-up and simulation. Hence, we added him to the coauthors, as it appears to be an essential question by the commenter and also by the editor, that went well beyond our original manuscript without a mathematician on board.

We expect our  $t_d$  estimates to be high estimates that help with the planning of the design of a ML that also quantifies drainage losses for more general applications than we aimed at (dry and drought conditions).

Going out of the topic of the paper under discussion we have to remind the commenter that we were unable to suggest a ML system that would also resolve the NRW inputs that could occur during rainfall events with an accuracy that is comparable to that what the proposed ML system can provide under dry conditions. There are four reason why this is a challenge of itself and would have to be solved by someone interested in these inputs at times when there are non-dry conditions in a separate study:

During rainfall events there is the issue of the splashing of droplets off the ground; a clear definition would be required of how to separate splash droplet water gains from other water gains; should this then be counted as NRW input (most likely not!), and how could it be distinguished from ground fog water inputs, condensation and adsorption?

Fog droplet inputs under rainfall conditions are also a difficult aspect that would require a special definition. Often during rainfall a near-surface layer of foggy air establishes. Via stable isotopes it can clearly be shown that the fog water comes from the concurrent rainfall, but once the rainfall stops the fog layer may still be present; should one now count fog droplet input during rainfall and after it has stopped both as NRW inputs? Or only the amount gained after rainfall has stopped? Or should both be considered a secondary pathway of normal rainfall input? In any case a more detailed assessment of the terminology would be required – which is not necessary if one focuses on drought conditions (knowing that by definition we do not count the first minutes and hours after a rainfall event a "drought").

During rainfall events the wetting of the outer side of the ML weighing pot becomes important. While one could quantify the water amount adsorbed to the outer side of the weighing pot in the laboratory, we would expect an increasingly higher weight of this water pool under real-world outdoor conditions because of e.g. algae growing on the outer face of the ML pot, and consequently also an increasing share of accumulated hydrophibic soil particles, etc.

No matter how one would choose the definitions for separating the components, the experimental errors to quantify "NRW" inputs during and (shortly) after rainfall events would explode. Currently we could only think of a system that would be 5–10 times less accurate than what we present for drought conditions.

This is not to say that it is impossible to achieve high-quality NRW input data during rainfall events, but it is to explain why we did not include this aspect in our project. By focusing on drought periods we circumvented all these technical challenges, well knowing that during rainfall events NRW inputs may not be perfectly zero (as our

approach assumes, see Eq. 1), hence our suggestion to explicitly mention that our estimates are conservative estimates if rainfall periods are included in the total. We added and changed in the discussion section to: "Under conditions with water lost via drainage, NRW inputs would be underestimated. Especially during and shortly after intensive rainfall periods, when drainage water flow is more likely (see Appendix F, Fig. F1 and Table F1), the application of the ML system is limited. During transition periods, shortly after rainfall, e.g. during nights when the sky clears after rainfall, NRW inputs may be underestimated. Therefore, we excluded such periods (see Eq. 1) from the analysis and limited our analysis for dry periods. Our longer-term NRW estimates might thus be conservative estimates if rainfall periods are included in the total hydrological input. At our site, drainage water flow from the ML pots reached low levels rather quickly after rainfall events (see the Appendix E and F for more details). Nevertheless, depending on soil characteristics and conditions, drainage water flow could persist for longer time (Fig. F1 and Table F1)." (lines 641–650).

We added some examples to explain the before used term "more universal" (the term was deleted in the new version): "However, to further develop this ML system and use it during and shortly after rainfall periods or to improve the measurements during other periods when the soil cannot hold excessive water universally, it is recommended to quantify drainage water flow." (lines 763–765).

Furthermore we added to the Abstract and to the Introduction: "Here we present a novel micro-lysimeter (ML) system and its application which allows to quantify very small water inputs from NRW **during rainfree periods** with an unprecedented high accuracy of  $\pm$  0.25 g, which corresponds to  $\pm$  0.005 mm water input." (lines 9–12) and "The goal of this study was to design and test an automated long-term ML system for NRW quantification to grasslands **during rainfree periods** in the field, that overcomes drawbacks of existing small ML systems in terms of hampered plant growth and altered canopy and soil temperatures compared to the control (surrounding area)." (lines 92–95).

Thus I don't understand why the authors showing in section 3.6 NRW inputs over one year here, when the know that the ML system cannot provide such data? I recommend to delete the section 3.6 as the shown total amounts are strongly biased by the inability of the ML system to correctly quantify NRW input during time were also drainage occurs.

For the analysis of the NRW input for one year we excluded periods when rainfall occurred as mentioned above. It is not the case that the ML system "cannot provide such data", but we make a conservative estimate that when there is rainfall and during the time after, when drainage losses from the ML must be expected, we set NRW input to 0 mm, thus we obtain a conservative estimate of an annual total NRW gain. We added the information that this is a conservative estimate: "Under conditions with water lost via drainage, NRW inputs would be underestimated. Especially during and shortly after intensive rainfall periods, when drainage water flow is more likely (see Appendix F, Fig. F1 and Table F1), the application of the ML system is limited. During transition periods, shortly after rainfall, e.g. during nights when the sky clears after rainfall, NRW inputs may be underestimated. Therefore, we excluded such periods (see Eq. 1) from the analysis and limited our analysis for dry periods. Our

longer-term NRW estimates might thus be conservative estimates if rainfall periods are included in the total hydrological input." (lines 641–648). We now explicitly mention in our second main objective: "design a ML system that allows differentiating between different NRW inputs, here defined as dew, hoar frost, fog, rime as well as water vapor adsorption events **during dry and drought conditions**". (lines 99–101). At our site, drainage water flow from the ML pots reached low levels rather quickly after rainfall events (see the Appendix E and Appendix F for more details). "Nevertheless, depending on soil characteristics and conditions, drainage water flow could persist for longer time (Fig. F1 and Table F1)." (lines 649–650).

please show the ML system installed in the field somewhere in the Material and Method section

We added photographs of the ML system and photographs taken during installation of the ML system to Appendix B.

5 please show day and nighttime here and show in a) more the just the very close vicinity of the ML system in the pictures.

We interpret that with "5" Fig. 5 was meant and not line 5 or section 5. We already show more than the close vicinity of the ML system in Fig. 5b. "To compare ML pot temperatures to temperatures of the surrounding, separate images were taken in a distance of ca. 100 cm (images not shown here) with a size of ca. 75x75 cm, to exclude potential influence of the ML on its approximate surrounding." (lines 368–370). We added these separate images for the commenter. Thermal images were taken during 18:27 and 05:15, no daytime images were taken. This ML system was designed to measure NRW inputs during nights

---

## Author Response (AR2)

**Comments to the author**:
Dear authors,

I read the revisions and your replies to the review reports and am satisfied with the improvements you made to the paper - both in the content (especially Appendix F), and in the organization of the paper.

Dear Editor,
Thank you for the fast handling of the manuscript and the helpful suggestions. Please find our answers below as blue indented text.

I have the following minor comments that I would ask you to take into consideration:

l. 42: Plant water status is not a scalar. It can therefore be changed, but not increased.

> We changed it to: "NRW inputs can increase the amount of water in plants (Limm et al., 2009; Munné-Bosch and Alegre, 1999) and change thereby the plant water status, which can lower plant water stress."

l. 93: Soils are unsaturated most of the time, otherwise they would not be soils.

> We deleted "unsaturated soils".

l. 234: Comma after completely?

> We added a comma after completely.

l. 354: A wet soil can be nearly or fully saturated. I have never seen 'heavily saturated' before.

> We changed it to: "fully saturated".

l. 519: When you fix the origin of a linear fit, the correlation coefficient looses its meaning (although I cannot easily explain why – but you have a mathematician among you)

> We agree, and the reason for not mentioning the $R^2$ in this context is due to the fact, that Pearson's linear correlation coefficient assumes normal distribution of both variables for which the correlation coeffecient is calculated, and when forcing a regression through the origin, the origin is mostly too far away from the data that the assumption that the data points (including the origin) ist not necessarily fullfilled. In our application the origin was close enough to report $R^2$ (but we agree that it shows an artificially high $R^2$ of 0.98 and we do not object to remove that information.

l. 911: volumic -> volumetric

> We changed it to: "volumetric".

l. 912: When a soil becomes unsaturated, the pressure head becomes the matric potential.

> We added: "The relation between the unsaturated hydraulic conductivity $k$, the volumetric water content $\theta$ and the pore-water pressure head $\psi$ (matrix potential) can be described by the following formula." In Zhan et al. 2016 they used "pore-water pressure head" and we picked up this terminology.

l. 913: You do not declare gamma until much later in the appendix. What do the asterisks of z and t signify?

> We added: "Where $\gamma$ is the slope angle (0° with our ML), $z_*$ is the axis perpendicular to the slope, and $t$ is time."
> $z$ is the vertical axis; the calculation was performed perpendicular to the surface and thus $z_*$ was used as surface-normal axis (or perpendicular to the slope).
> The asterisk of the "t" was a wrong punctuation mark. We deleted it.

l. 916 and below: I think the air-entry value should be subtracted from the matric potential. You want the term exp(psi – psi_sub_ae) to be equal to one at the air-entry value.
> We agree and that is also how we modeled the drainage flow. In Eq. (F6) the air entry point is subtracted from the matric potential (in the denominator), thus in line with what the Editor expects.

Appendix F. You switch fonts at some point and use upper case Greek. From the text it appears this is not intentional.
> Thank you, we corrected it.

When you have addressed these, the paper shoud be ready for publication.

Yours sincerely,

Gerrit de Rooij
Editor